# Biological brain age prediction using machine learning on structural neuroimaging data: Multi-cohort validation against biomarkers of Alzheimer's disease and neurodegeneration stratified by sex

Irene Cumplido-Mayoral[1,2], Marina García-Prat[1], Grégory Operto[1,3,4], Carles Falcon[1,3,5], Mahnaz Shekari[1,2,3], Raffaele Cacciaglia[1,3,4], Marta Milà-Alomà[1,2,3,4], Luigi Lorenzini[6], Silvia Ingala[6], Alle Meije Wink[6], Henk JMM Mutsaerts[6], Carolina Minguillón[1,3,4], Karine Fauria[1,4], José Luis Molinuevo[1†], Sven Haller[7], Gael Chetelat[8], Adam Waldman[9], Adam J Schwarz[10], Frederik Barkhof[6,11], Ivonne Suridjan[12], Gwendlyn Kollmorgen[13], Anna Bayfield[13], Henrik Zetterberg[14,15,16,17,18], Kaj Blennow[14,15], Marc Suárez-Calvet[1,3,4,19], Verónica Vilaplana[20]*, Juan Domingo Gispert[1,3,5]*, ALFA study, EPAD study, ADNI study, OASIS study

[1]Barcelonaβeta Brain Research Center, Pasqual Maragall Foundation, Barcelona, Spain; [2]Universitat Pompeu Fabra, Barcelona, Spain; [3]IMIM (Hospital del Mar Medical Research Institute), Barcelona, Spain; [4]CIBER Fragilidad y Envejecimiento Saludable (CIBERFES), Madrid, France; [5]Centro de Investigación Biomédica en Red de Bioingeniería, Biomateriales y Nanomedicina (CIBER-BBN), Madrid, Spain; [6]Department of Radiology and Nuclear Medicine, Amsterdam Neuroscience, Vrije Universiteit Amsterdam, Amsterdam, Netherlands; [7]CIRD Centre d'Imagerie Rive Droite, Geneva, Switzerland; [8]Normandie Univ, UNICAEN, INSERM, U1237, PhIND "Physiopathology and Imaging of Neurological Disorders", Institut Blood and Brain, Cyceron, France; [9]Centre for Dementia Prevention, Edinburgh Imaging, and UK Dementia Research Institute at The University of Edinburgh, Edinburgh, United Kingdom; [10]Takeda Pharmaceutical Company Ltd, Cambridge, United States; [11]Institutes of Neurology and Healthcare Engineering, University College London, London, United Kingdom; [12]Roche Diagnostics International Ltd, Rotkreuz, Switzerland; [13]Roche Diagnostics GmbH, Penzberg, Germany; [14]Institute of Neuroscience and Physiology, University of Gothenburg, Mölndal, Sweden; [15]Clinical Neurochemistry Laboratory, Sahlgrenska University Hospital, Mölndal, Sweden; [16]Department of Neurodegenerative Disease, UCL Queen Square Institute of Neurology, London, United Kingdom; [17]Hong Kong Center for Neurodegenerative Diseases, Hong Kong, China; [18]UK Dementia Research Institute at UCL, London, United Kingdom; [19]Servei de Neurologia, Hospital del Mar, Barcelona, Spain; [20]Department of Signal Theory and Communications, Universitat Politècnica de Catalunya, Barcelona, Spain

*For correspondence:
veronica.vilaplana@upc.edu (VV);
jdgispert@barcelonabeta.org
(JDG)

Present address: †H.Lundbeck
A/S, Copenhagen, Denmark

**Abstract** Brain-age can be inferred from structural neuroimaging and compared to chronological age (brain-age delta) as a marker of biological brain aging. Accelerated aging has been found in neurodegenerative disorders like Alzheimer's disease (AD), but its validation against markers of neurodegeneration and AD is lacking. Here, imaging-derived measures from the UK Biobank dataset (N=22,661) were used to predict brain-age in 2,314 cognitively unimpaired (CU) individuals at higher risk of AD and mild cognitive impaired (MCI) patients from four independent cohorts with available biomarker data: ALFA+, ADNI, EPAD, and OASIS. Brain-age delta was associated with abnormal amyloid-β, more advanced stages (AT) of AD pathology and *APOE*-ε4 status. Brain-age delta was positively associated with plasma neurofilament light, a marker of neurodegeneration, and sex differences in the brain effects of this marker were found. These results validate brain-age delta as a non-invasive marker of biological brain aging in non-demented individuals with abnormal levels of biomarkers of AD and axonal injury.

## Editor's evaluation

The study has significance for the field of dementia research and neurodegenerative diseases more broadly. Using the brain-age paradigm, the main findings are that having an older-appearing brain is associated with more advanced stages of amyloid and tau pathology, higher white matter hyperintensities, higher plasma NfL and carrying the APOE-e34 allele. Findings were broadly similar in cognitively normal people and people with mild cognitive impairment and there is also some evidence for sex differences.

## Introduction

Age is the main risk factor for Alzheimer's Disease (AD) and most neurodegenerative diseases. However, the mechanisms underlying this association are still poorly understood (*Fjell et al., 2014*). Both normal aging and AD are associated with region-specific cerebral morphological changes characterized by the occurrence of atrophy (*Bakkour et al., 2013*; *Fjell et al., 2014*). Both aging and AD have differential and partially overlapping effects on specific regions of the cerebral cortex like, for instance, the dorsolateral prefrontal cortex (*Bakkour et al., 2013*; *Fjell et al., 2014*; *Pichet Binette et al., 2020*). Conversely, some regions are predominantly affected by age (e.g. calcarine cortex) and some others are predominantly affected by AD (e.g. medial temporal cortex; *Bakkour et al., 2013*). A better understanding of the mechanistic links between the brain aging process and neurodegenerative diseases is an urgent priority to develop effective strategies to deal with their rising burden amid an ageing population (*Franke and Gaser, 2019*). Therefore, a growing amount of research is focusing on using neuroimaging techniques to develop a biomarker of biological brain aging. In this framework, the concept of brain-age has emerged as an appealing comprehensive marker that enables determining on an individual basis, the risk for age-associated brain diseases (*Cole and Franke, 2017a*; *Cole et al., 2017c*; *Franke et al., 2010*; *Franke and Gaser, 2019*). However, this is a challenging task because, even though the cerebral structural changes related to aging are well established, the older population is characterized by substantial variation in neurobiological aging trajectories (*Cole et al., 2018*; *Fjell et al., 2014*).

Recently, machine learning techniques have gained popularity as brain-age prediction models (*Cole et al., 2017b*; *Dafflon et al., 2020*; *de Lange et al., 2019*; *Franke and Gaser, 2019*), due to their ability in identifying relevant data-driven patterns within complex data (*Zhavoronkov et al., 2019*). These models learn the association between chronological age and cerebral morphological features derived from structural magnetic resonance imaging (MRI) in healthy individuals, yielding a predicted brain-age for each individual. Individuals with a predicted brain-age higher than their chronological age may have an 'older' brain than expected, whereas an individual with an estimated brain-age lower than their chronological age has a 'younger' brain. Subtracting chronological age from estimated brain-age hence provides an estimate of accelerated brain aging, namely the brain-age delta. Recent literature has shown the adequacy of using a brain-age predicted measurement in the assessment of the clinical severity of AD, by finding higher brain-age deltas in AD and individuals with mild cognitive impairment (MCI) with respect to cognitively unimpaired (CU) individuals (*Beheshti et al., 2018*;

*Kaufmann et al., 2019*). A higher brain-age delta has also been reported in other diseases, such as multiple sclerosis, epilepsy and psychiatric disorders, with respect to healthy controls (*Beheshti et al., 2018*; *Kaufmann et al., 2019*). In addition, brain-age delta has also been associated with other biological measures such as: lifestyle factors (*Cole, 2020*)**,** cognition (*Beheshti et al., 2018*; *Cole, 2020*) hypertension (*de Lange et al., 2020a*) and prediction of mortality (*Cole et al., 2018*).

Even though these studies support the association of brain-age delta as a biomarker of biological aging with relevance to various brain diseases, it is of interest to further validate this measurement in association with specific biological markers of AD pathology (i.e. Amyloid-β[Aβ] and tau pathology), neurodegeneration and cerebrovascular disease in the earliest stages of AD. This is a very relevant aspect since the recent AD research framework criteria defines AD as a biological construct, namely the presence of both abnormal Aβ (A+) and tau (T+) biomarkers, regardless of clinical manifestations (*Jack et al., 2018*). The term 'Alzheimer's pathological change' is proposed whenever there is evidence of Aβ but not tau pathology (A+T-). The umbrella term 'Alzheimer's *continuum*' includes both 'Alzheimer's pathological change' (A+T-) and 'Alzheimer's Disease' (A+T + ). Under this definition, A-T +individuals would not fall into the AD *continuum*. Then, under this framework, neurodegeneration biomarkers (N) and cognitive status (i.e. CU, MCI and dementia syndromes) are used to stage disease progression.

Recent literature has studied the associations between brain-age delta and the above-mentioned biomarkers with different aims. Brain-age delta was recently associated with plasma biomarkers of neurodegeneration and with imaging biomarkers of cerebrovascular disease in the 1946 British Birth Cohort *Wagen et al., 2022*; however, it was not associated with CSF biomarkers of neurodegeneration (*Millar et al., 2022*). Moreover, although brain-age delta was significantly associated with AD biomarkers of amyloid and tau in MCI individuals, these associations were not found in CU individuals (*Millar et al., 2022*; *Wagen et al., 2022*). Conversely, Another study focusing on the impact of training the brain-age prediction model in individuals with Aβ pathology (Aβ+) showed that CU Aβ+individuals had a higher brain-age delta than CU Aβ- individuals (*Ly et al., 2020*). Other lines of research have employed brain-age delta to predict conversion to different disease stages. For instance, it was previously studied the impact of using brain-age, alone and in combination with several biomarkers, to predict progression from MCI to AD (*Popescu et al., 2020*). In addition, a recent study used brain-age measurements to identify amnestic MCI (aMCI), the typical clinical presentation of prodromal AD, from other individuals with MCI, by studying the association with AD risk factors such as apolipoprotein E (*APOE*) and Aβ (*Huang et al., 2021*). The association of brain-age with these biomarkers have also been shown in other diseases, by which brain-age was associated with Aβ deposition in Down syndrome (*Cole et al., 2017b*). Nonetheless, there remains a need to replicate some of these results and to study the associations between brain-age prediction and abnormal biomarkers of AD and neurodegeneration in preclinical and prodromal AD stages in different and independent cohorts and in a larger sample size. These results would be particularly informative to potentially inform therapeutic interventions.

Moreover, given that female individuals have a higher AD prevalence compared to males (*Nebel et al., 2018*) and display different lifetime trajectories in the brain morphological features (*Gennatas et al., 2017*), it is of interest to determine the effect of sex on brain age delta and its interaction with AD biomarkers. Literature describes sex differences in AD biomarkers, such as that females with abnormal Aβ who are *APOE*-ε4 carriers show greater subsequent increase in cerebrospinal fluid (CSF) tau than their male counterparts (*Buckley et al., 2019*), or that females with higher Aβ burden show higher entorhinal cortical tau than their male counterparts (*Buckley et al., 2019*). Conversely, levels of the neurodegeneration biomarker CSF neurofilament light (NfL) have been widely reported to be higher in males than in females (*Mielke, 2020*; *Milà-Alomà et al., 2020*). In line with this, sex differences in the brain-age delta have been also reported. For instance, AD risk factors have been associated with greater brain aging in women than men (*Sanford et al., 2022*; *Subramaniapillai et al., 2021*) and a lower brain aging has been found in the lower in the prefrontal cortex and insula in females (*Sanford et al., 2022*). These results need to be replicated and further studied in different cohorts.

Therefore, in the present study, we aim to validate brain-age delta as a clinically relevant marker of brain aging, which can be impacted by AD pathology and neurodegeneration even in non-demented individuals. For this purpose, we determine the association between the predicted structural brain-age

delta with biomarkers and risk factors for AD and neurodegeneration in non-demented individuals, as well as to study the effect of sex on these associations. We trained a model to predict the brain-age separately for females and males, using machine learning on imaging-derived measures of cortical thickness, cortical volume, and subcortical volume from the UK Biobank cohort (N=22,661). Using this model, we then estimated brain-age in four independent cohorts: ALFA+ (N=380), ADNI (N=719), EPAD (N=808), and OASIS (N=407). In each cohort, we studied the associations of brain-age delta with biomarkers of AD pathology (CSF Aβ and p-tau as continuous values, as well as categorized in AT stages), the *APOE*-ε4 genotype which is the main genetic risk factor for AD, neurodegeneration (CSF and plasma NfL), and small vessel disease (White Matter Hyperintensities [WMH]). Finally, we studied the sex differences in brain age prediction and the sex effects with these biomarkers on brain-age delta.

## Results

### Participants' characteristics

*Table 1* summarizes the demographic characteristics of the cohorts included in the study. ADNI and EPAD cohorts included both CU and MCI individuals, while the UK Biobank, ALFA + and OASIS cohorts only included CU individuals. *Table 2* summarizes the variables used to study the associations with brain-age delta, which included biomarkers for AD (Aβ positron emission tomography [PET] and CSF Aβ and p-tau), neurodegeneration (CSF and plasma NfL), and cerebrovascular pathology (WMH on MRI), as well as the aging signature composite (*Bakkour et al., 2013*), both cross-sectional and longitudinally. The aging signature composite is a map of specific brain regions that undergo cortical thinning in normal aging, which has been used as a proxy measurement for brain aging. These validation variables were correlated with chronological age for all cohorts (see *Appendix 1—table 1*). Some of the participants for ALFA+ (N=25), ADNI (N=116), and EPAD (N=71) fell into the A-T +group, corresponding to non-AD pathologic change. Since our aim was to specifically validate the brain-age delta measurements in the AD *continuum*, we excluded these participants from subsequent analyses; and they are reported within *Table 1* and *Table 2* solely for descriptive purposes. In addition, the number of MCI individuals with available data of CSF NfL and of aging signature change was relatively low and, therefore, these variables were excluded from the analysis in MCI individuals.

### Brain-age prediction and chronological age

We trained the prediction model using the UK Biobank cohort and tested the model using four independent cohorts (ALFA+, ADNI, EPAD, and OASIS), as shown in *Figure 1*. *Table 3* shows the prediction accuracy for the combined female and male predictions. The average prediction accuracy of the model run on UK Biobank using 10-fold cross-validation as measured by the mean absolute error (MAE) and by Pearson's correlation were MAE = 4.19 and *R*=0.71 (*Table 3* and *Figure 1—figure supplement 1*).

We then investigated the association of predicted brain-age with chronological age on each of the independent cohorts. All the cohorts showed a similar positive correlation and fitting performance metrics as measured by the MAE, R and root mean squared error (RMSE) between chronological age and predicted brain-age. Correlation coefficients were not different between cohorts (*P*>0.05, for all comparisons, see *Appendix 1—table 2*). We also studied the performance metrics for the two different diagnostic groups (CU and MCI) for each cohort, see *Appendix 1—table 3*.

In order to study the effect of sex on brain age prediction, we also computed the performance metrics stratified by females and males (*Appendix 1—table 4* and *Appendix 1—table 5*). Correlations and fitting performance metrics were not significantly different between females and males (Pearson's r (William's test), *P*>0.05; RMSE (F-test) *P*>0.05), see *Appendix 1—table 6*. Plots of the correlations between predicted brain-age and chronological age for females and males in each of the cohorts can be seen in the *Figure 1—figure supplement 2*. Additionally, results for a secondary analysis in which we compared the fitting performance of XGBoost brain-age and of the neuroanatomical aging signature (*Bakkour et al., 2013*) with respect to chronological age can be seen in *Appendix 1—table 7*.

**Table 1.** Sample demographics and characteristics separated by cohort and by diagnosis.

| Characteristics | CU | | | | | MCI | |
| --- | --- | --- | --- | --- | --- | --- | --- |
| | UK Biobank | ALFA+ | ADNI | EPAD | OASIS | ADNI | EPAD |
| | (N=22,661) | (N=380) | (N=284) | (N=653) | (N=407) | (N=435) | (N=155) |
| Age, years | 64.54 (7.55) | 60.61 (4.72) | 71.42 (6.36) | 64.96 (7.01) | 69.07 (9.42) | 71.09 (7.31) | 69.08 (6.97) |
| Age range, years | [44, 81] | [48, 73] | [55, 89] | [50, 88] | [42, 89] | [55, 91] | [52, 88] |
| Female, n (%) | 11,767 (51.92) | 254 (60.76) | 126 (50.00) | 386 (59.11) | 244 (59.95) | 249 (50.00) | 81 (47.74) |
| Education, years | 17.75 (5.42) | 13.43 (3.71) | 16.54 (2.49) | 14.83 (3.56) | 15.93 (2.59) | 16.23 (2.71) | 14.17 (3.77) |
| APOE-ε4 carriers, n (%) | 6,334 (27.95) | 221 (52.87) | 72 (28.57) | 217 (33.23) | 118 (28.99) | 218 (43.78) | 60 (38.71) |
| MMSE | - | 29.15 (0.95) | 28.985 (1.24) | 28.82 (1.40) | 29.03 (1.31) | 27.57 (2.19) | 27.86 (1.97) |

Notes: Data are expressed as mean (M) and standard deviation (SD) or percentage (%), as appropriate. Abbreviations: APOE, apolipoprotein E; MMSE, Mini-Mental State Examination.

**Table 2.** Biomarkers separated by cohort and by diagnosis.

| | CU | | | | | | | | MCI | | | |
| BIOMARKERS | ALFA+ | | ADNI | | EPAD | | OASIS | | ADNI | | EPAD | |
| | N | Mean (SD) | N | Mean (SD) | N | Mean (SD) | N | Mean (SD) | N | Mean (SD) | N | Mean (SD) |
|---|---|---|---|---|---|---|---|---|---|---|---|---|
| Centiloids | 0 | - | 0 | - | 0 | - | 407 | 13.468 (28.138) | 0 | - | 0 | - |
| CSF Aβ42 (pg/mL)* | 380 | 1318.059 (599.223) | 284 | 1223.890 (556.648) | 653 | 1403.617 (681.736) | 0 | - | 435 | 986.248 (446.402) | 155 | 1245.181 (741.756) |
| CSF p-tau (pg/mL) | 380 | 16.289 (7.813) | 283 | 22.234 (9.692) | 627 | 18.326 (8.380) | 0 | - | 434 | 26.490 (14.402) | 151 | 24.715 (14.897) |
| CSF NfL (pg/mL)† | 380 | 82.717 (29.124) | 26 | 1052.444 (376.095) | 0 | - | 0 | - | 48 | 1383.638 (918.231) | 0 | - |
| Plasma NfL (pg/mL) | 368 | 10.519 (3.739) | 184 | 35.843 (17.988) | 0 | - | 0 | - | 404 | 38.157 (18.908) | 0 | - |
| WMH volume | 360 | 0.045 (0.845) | 240 | -0.0085 (1.267) | 456 | 0.038 (1.072) | 0 | - | 458 | -0.005 (1.229) | 108 | 0.048 (1.076) |
| Aging signature† | 360 | 2.387 (0.071) | 240 | 2.284 (0.105) | 456 | | 0 | - | 458 | 2.251 (0.109) | 0 | - |
| Aging signature V2† | 187 | 2.376 (0.072) | 45 | 2.299 (0.118) | 0 | - | 0 | - | 46 | 2.257 (0.119) | 0 | - |
| Aging signature change ($\frac{V2-V1}{t}$)† | 187 | -0.003 (0.011) | 45 | -0.0007 (0.037) | 0 | - | 0 | - | 46 | -0.003 (0.050) | 0 | - |

Data are expressed as mean (M) and standard deviation (SD) or percentage (%), as appropriate. Amyloid-β status was defined by CSF (ALFA+, ADNI and EPAD) or amyloid PET (OASIS). For ALFA+ and ADNI, we calculated the aging signature from MRI scans acquired 3 years later than the original MRI scan, called aging signature V2. Aging signature change was calculated as the difference in aging signature over these two MRI scans.

Abbreviations: CSF, cerebrospinal fluid; NfL, neurofilament light; WMH, White Matter Hyperintensities.

*Individuals that fell into the A-T+group: 25 from ALFA+, 116 from ADNI and 71 from EPAD.

†As the number of MCI individuals with CSF NfL and aging signature change was relatively low, we excluded them from the following results.

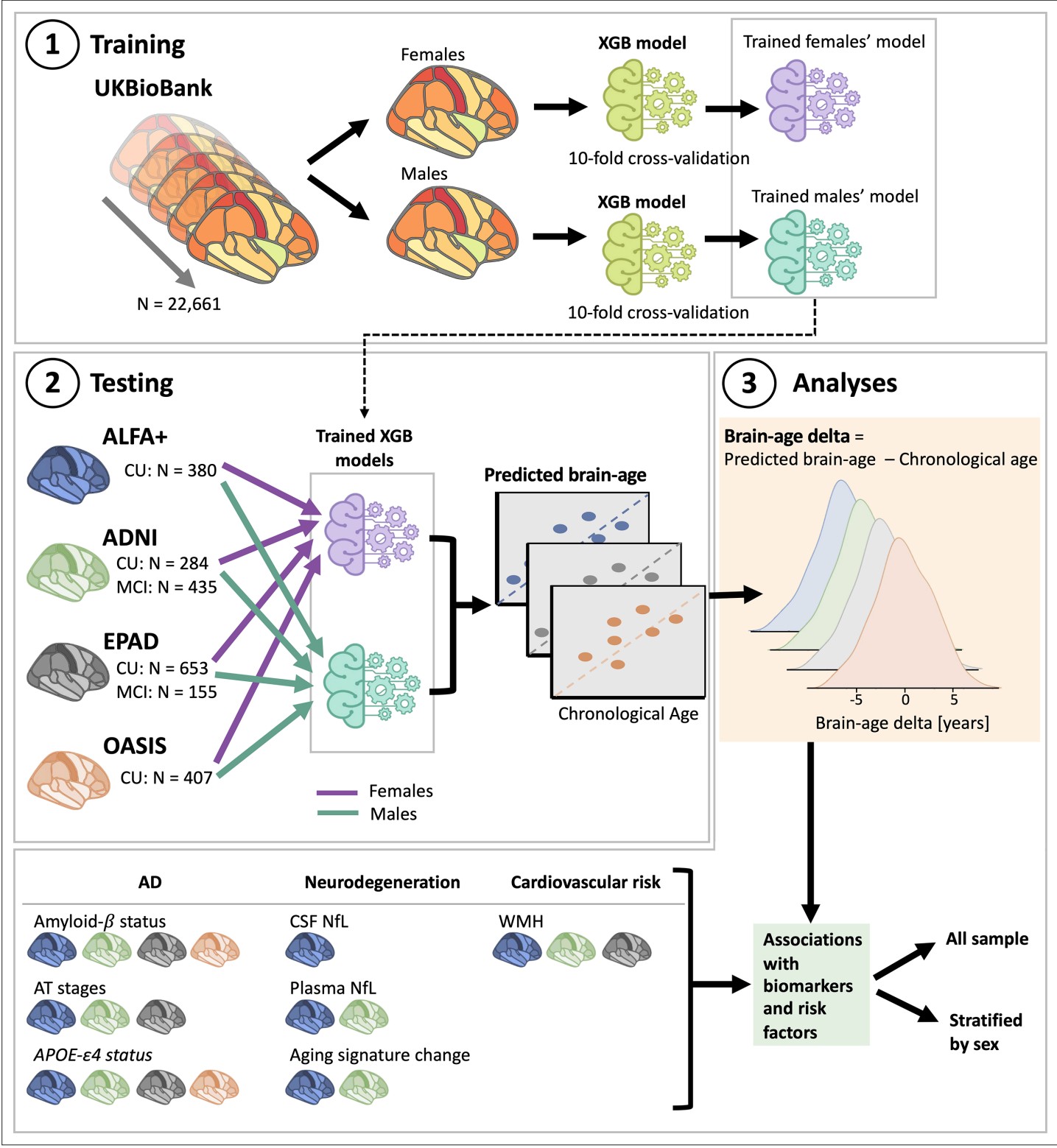

**Figure 1.** Overview of project steps. Illustration of the methods used to generate predicted brain-age and to study the associations between the brain-age delta and the biomarkers and risk factors used. 3D T1-weighted MRI scans across all cohorts were segmented into volumes and thickness using the Desikan-Killiany and the aseg atlas. **1. Training phase**: We trained XGBoost regressor models for females and males from the UK Biobank. For this we performed a cross-validation scheme with 10-folds and 10 repeats per fold. **2. Testing phase**: We tested the age prediction models on unseen data from independent cohorts: ALFA+ (in blue), ADNI (in green), EPAD (in gray), and OASIS (in orange). **3 Analyses phase**: We computed the brain-age delta for each cohort. We then studied the associations with the biomarkers and risk factors of AD, neurodegeneration, and cardiovascular risk. We performed

*Figure 1 continued on next page*

*Figure 1 continued*

these analyses within the whole sample and stratified by sex. The table on the bottom left shows the available biomarkers and risk factor available for which cohorts.

The online version of this article includes the following figure supplement(s) for figure 1:

**Figure supplement 1.** Predicted brain-age in validation subsamples of UK Biobank for females (blue) and males (orange).

**Figure supplement 2.** Predicted brain-age versus chronological age for (**a**) ALFA+, (**b**) ADNI, (**c**) EPAD, and (**d**) OASIS cohorts.

## Brain regions associated with aging

We computed the SHapley Additive exPlanation (SHAP) values, which reflect the marginal contribution of each brain region to the brain-age prediction, using the UK Biobank dataset. SHAP values interpret the impact in the prediction of the values of volume or cortical thickness for a given brain region. In other words, they reflect the most important features that consistently influenced the prediction of brain-age and whether the decrease or increase of each region impacted into predicting a higher or lower brain-age. The SHAP values were computed separately for females and males. We compared the regions with higher SHAP values for females and males, and vice-versa, by averaging the SHAP values within each sex separately and then subtracting the mean SHAP of males to the mean SHAP of females.

There were regions whose SHAP values were high in both females and males, including the volumes of the amygdala, nucleus accumbens, cerebellar white matter, lateral ventricles and the insula, as well as the cortical thickness of the superior-temporal cortex. All the brain regions with consistent highest SHAP values for females and males are shown in *Figure 2a–b*, as well as the effect of each region (larger or lower value) on predicting a higher brain-age. Conversely, the thickness of regions such as the transverse temporal cortex, the pars triangularis, the inferior parietal cortex and the left frontal pole thickness, as well as the volume of the left entorhinal cortex had higher SHAP values in females than in males, while the opposite occurred with the thickness of the left isthmus cingulate and the right cuneus and the cortical volume of the superior frontal and right rostral middle regions (*Figure 2c*).

In *Figure 2d*, we can see the fit of three example regions whose SHAP values were different for females and males against chronological age. For example, the bi-lateral superior frontal volumes decreased more over the years within males than females. This result was seen as the interaction of sex with age ($P_{interaction}$ <0.001). We also found an interaction effect of sex and age for the isthmus cingulate thickness ($P_{interatcion}$ <0.001), by which the thickness of males decreased more over the years than from the females. On the contrary, we also found regions, such as the middle temporal thickness, whose slope was not different between for both sexes ($P_{interaction}$ = 0.671), but which appeared to be lower for females than for males. All the results of this exploratory analysis can be seen in *Appendix 1—table 8*.

## Associations with AD biomarkers and risk factors

We studied the association between brain-age delta and AD biomarker classifications (Aβ status, AT stages) and *APOE*-ε4 status in all the independent cohorts pooled together, with a linear model adjusting for the effect of age and sex (*Figure 3* and *Table 4*). Aβ status was defined by CSF (ALFA+, ADNI, and EPAD) or amyloid PET (OASIS) using pre-established cut-off values (*Hansson et al., 2018*; *Milà-Alomà et al., 2020*; *Salvadó et al., 2019*; *Schindler et al., 2018*). Brain-age delta was higher in MCI with respect to CU individuals ($P_{FDR}$ <0.001). In both CU and MCI, a higher brain-age delta was significantly associated with abnormal Aβ status (CU: $P_{FDR}$ <0.001 and MCI: $P_{FDR}$ <0.001) and with progressive AT stages (CU: $P_{FDR}$ <0.001 and MCI: $P_{FDR}$ <0.001) (see *Table 4* and *Appendix 1—table 9* for more details). The mean brain-age delta values for the different Aβ status and AT stages can be found in *Appendix 1—table 10*. The brain-age effect on AT stages was progressive, as that of the A+T- group was larger than that of A-T-, while the brain-age delta of A+T + was larger than those of the other two previous stages (*Table 4* and *Figure 3a*). Brain-age delta was also significantly associated with *APOE* status (CU: $P_{FDR}$ = 0.002 and MCI: P=0.017, $P_{FDR}$ = 0.040). In particular, *APOE*-ε4 carriers had larger brain-age deltas (i.e. older-appearing brains than expected for their chronological age) compared to *APOE*-ε33 individuals for both CU ($\beta$=0.173, $P_{FDR}$ = 0.003) and MCI ($\beta$=0.273, $P_{FDR}$ = 0.008; see *Table 4* and *Figure 3*). The mean brain-age delta values for the different *APOE* status can be found in *Appendix 1—table 7*. These results were consistent with the results from the within-cohort analyses (see *Appendix 1—table 11*).

**Table 3.** Prediction metrics for all independent cohorts.

| Cohorts | Correlation with age | | MAE (y) | R² | RMSE |
|---|---|---|---|---|---|
| | R | P-value | | | |
| Before bias correction | | | | | |
| UK Biobank | 0.712 (0.007) | <0.001 | 4.19 (0.07) | 0.51 (0.03) | 5.25 (0.08) |
| ALFA+ | 0.448 | <0.001 | 4.31 | 0.20 | 4.18 |
| ADNI | 0.587 | <0.001 | 7.21 | 0.34 | 5.47 |
| EPAD | 0.629 | <0.001 | 4.63 | 0.40 | 5.62 |
| OASIS | 0.733 | <0.001 | 6.99 | 0.54 | 6.42 |

The Pearson's correlation coefficient (R) between predicted brain-age and chronological age, R², root mean square error (RMSE), and mean absolute error (MAE) for UK Biobank and for each of the independent cohorts before bias correction. For UK Biobank, the metrics, given as mean (standard deviation) are computed from 10-fold cross validation repeated 10 times.

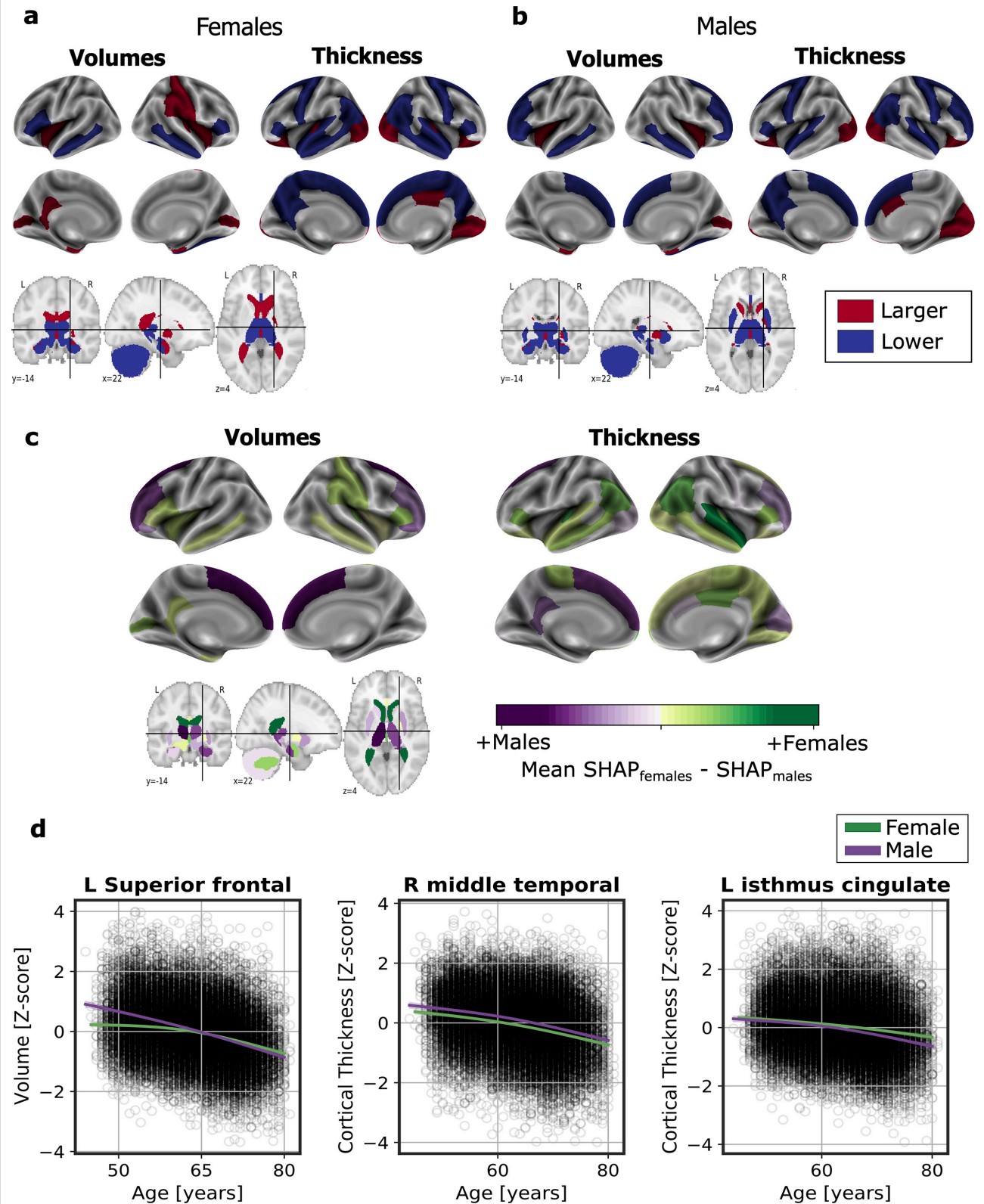

**Figure 2.** Significant SHAP-selected brain regions most important in prediction for (**a**) females and (**b**) males separately. Significance was studied by assessing the stability of the region's importance by performing subsampling of data over 1,000 permutations. Colored regions had a $P < 0.05$ corrected for multiple comparisons using Bonferroni correction approach. Regions in red show larger volume or cortical thickness, while regions in blue show lower volume or cortical thickness. In (**c**), comparison for the regions with higher SHAP values that were significant for females (green) and

*Figure 2 continued on next page*

*Figure 2 continued*

males (purple). The color map shows the results from subtracting the males' mean SHAP value to the female's mean SHAP value for each region. In (**d**), examples of the fit of three significant SHAP-selected regions against chronological age for females and males. For visualization purposes, nonparametric smoothing spline functions were used to fit the data (mean ± 95%CI).

We next studied the association between brain-age delta and AD biomarkers and risk factors stratified by sex (*Table 5*). In general, the same associations found with the whole sample was seen for females and males separately. However, although a higher brain-age delta was significantly associated with progressive AT stages both for females (CU: $P_{FDR}$ = 0.002 and MCI: $P_{FDR}$ <0.001) and males (CU: $P_{FDR}$ = 0.019 and MCI: $P_{FDR}$ <0.001), brain-age delta of A+T + was significantly larger than those of the other two previous stages (A-T- and A+T-) in CU females ($\beta$=0.431, $P_{FDR}$ = 0.008) but not in CU males ($\beta$=0.139, $P_{FDR}$ = 0.424). Brain-age delta was only significantly associated with *APOE* status in CU males ($P_{FDR}$ = 0.028), while this association was not seen in the rest of subgroups (CU and MCI females and MCI males). The rest of the associations tested with AD biomarkers, such as the interactions with sex, not reported here were non-significant.

## Associations with neurodegeneration biomarkers

We next tested the associations between brain-age delta and neurodegeneration biomarkers (*Figure 3* and *Table 4*). CSF NfL, plasma NfL and longitudinal change of the aging signature were available in ALFA + and ADNI. The positive associations between brain-age deltas and plasma NfL were significant within the CU ($\beta$=0.154, $P_{FDR}$ = 0.003) and MCI individuals ($\beta$=242, $P_{FDR}$ <0.001). CSF NfL was not significantly associated with brain-age delta ($\beta$=0.079, $P_{FDR}$ = 0.209). The association between the longitudinal change in the aging signature composite, which was obtained from the cortical thickness in aging-vulnerable regions, and brain-age delta was not statistically significant ($\beta$=0.053, $P_{FDR}$ = 0.415).

We next studied the association between brain-age delta and neurodegeneration biomarkers stratified by sex (*Table 5*). The associations between brain-age delta and plasma NfL were significant within the CU and MCI females (CU: $\beta$=0.191, $P_{FDR}$ = 0.003 and MCI: $\beta$=0.342, $P_{FDR}$ = 0.003), but not within the males (CU: $\beta$=0.110, $P_{FDR}$ = 0.217 and MCI: $\beta$=0.141, $P_{FDR}$ = 0.137). However, the interaction

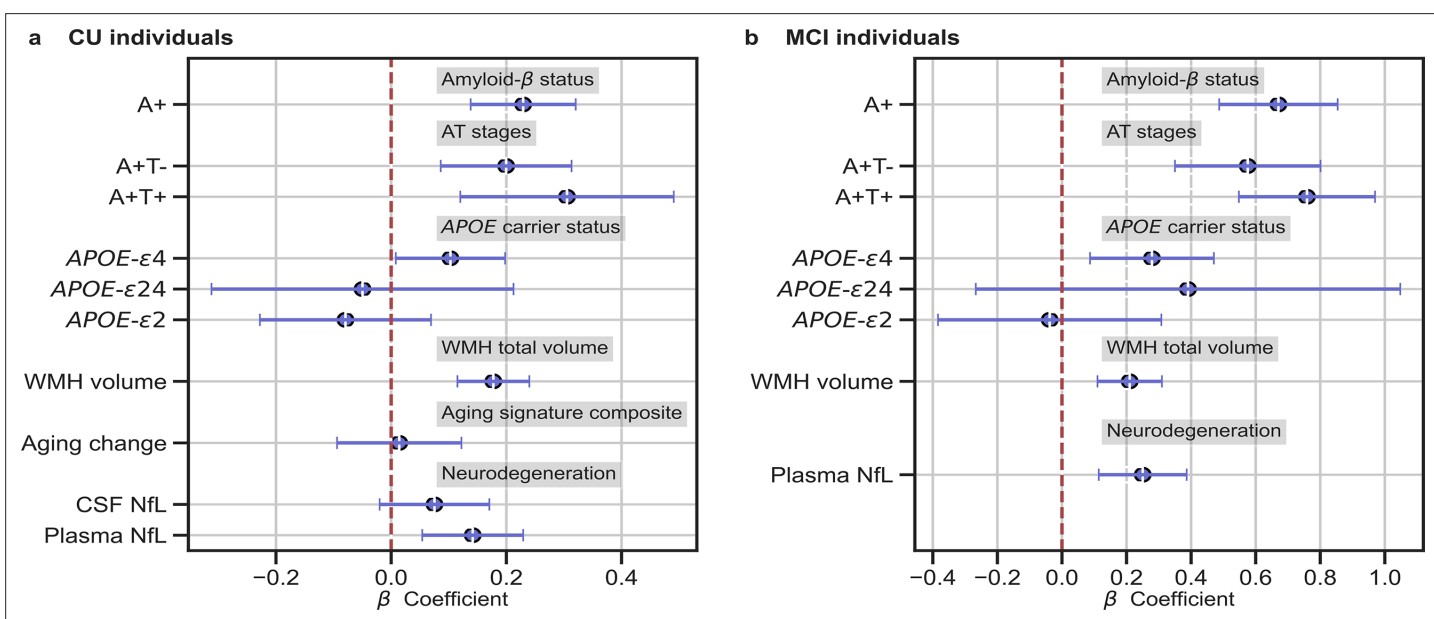

**Figure 3.** In (**a**) and (**b**), the standardized associations ($\beta$±95% CI) between measures of brain-age delta validation variables for (**a**) CU individuals and (**b**) MCI individuals. Variables include AD biomarkers and risk factors: amyloid-β status, AT stages and APOE status; and neurodegeneration markers (available in ALFA + and ADNI): CSF NfL, plasma NfL and aging signature change. The analyses included age and sex as covariates. Sample size for each variable can be seen in *Table 4*.

**Table 4.** Relationships between validation variables and brain-age delta for all CU and MCI individuals.

| Model | | β | SE | P-Value | [0.025] | [0.975] | N | Effect size | FDR corr P-Value |
|---|---|---|---|---|---|---|---|---|---|
| **CU Individuals** | | | | | | | | | |
| Amyloid-β pathology (ref: A-) | | 0.234 | 0.047 | **<0.001** | 0.140 | 0.325 | 1634 | 0.222 | **<0.001** |
| Amyloid-β / Tau pathology (ref: A-T-) | A+T- | 0.2023 | 0.059 | **<0.001** | 0.015 | 0.394 | 1162 | 0.275 | **<0.001** |
| | A+T + | 0.310 | 0.096 | **0.003** | 0.101 | 0.500 | | 0.300 | **0.008** |
| *APOE* status (ref: *APOE*-ε33) | *APOE*-ε2 | –0.124 | 0.082 | 0.130 | –0.321 | 0.334 | 1634 | 0.122 | 0.227 |
| | *APOE*-ε4 | 0.173 | 0.052 | **0.001** | 0.071 | 0.274 | | 0.172 | **0.003** |
| | *APOE*-ε24 | 0.012 | 0.144 | 0.936 | –0.271 | 0.294 | | 0.011 | 0.999 |
| WMH volume[*] | | 0.160 | 0.030 | **<0.001** | 0.111 | 0.231 | 972 | 0.028 | **<0.001** |
| CSF NfL[†] | | 0.079 | 0.049 | 0.112 | –0.019 | 0.176 | 378 | 0.006 | 0.209 |
| Plasma NfL[†] | | 0.154 | 0.045 | **0.001** | 0.066 | 0.242 | 508 | 0.024 | **0.003** |
| Brain Atrophy[†] | | 0.053 | 0.048 | 0.272 | –0.041 | 0.146 | 152 | 0.003 | 0.415 |
| **MCI Individuals** | | | | | | | | | |
| Amyloid-β pathology | | 0.640 | 0.089 | **<0.001** | 0.465 | 0.816 | 218 | 0.665 | **<0.001** |
| Amyloid-β / Tau pathology (ref: A-T-) | A+T- | 0.564 | 0.109 | **<0.001** | 0.350 | 0.778 | 218 | 0.592 | **<0.001** |
| | A+T + | 0.720 | 0.102 | **<0.001** | 0.519 | 0.920 | | 0.720 | **<0.001** |
| *APOE* status (ref: *APOE*-ε33) | *APOE*-ε2 | 0.007 | 0.167 | 0.968 | –0.273 | 0.978 | 218 | 0.001 | 0.999 |
| | *APOE*-ε4 | 0.273 | 0.093 | **0.003** | 0.091 | 0.456 | | 0.281 | **0.008** |
| | *APOE*-ε24 | 0.352 | 0.319 | 0.269 | –0.273 | 0.978 | | 0.359 | 0.415 |
| WMH volume | | 0.222 | 0.054 | **<0.001** | 0.117 | 0.327 | 191 | 0.040 | **<0.001** |
| Plasma NfL[‡] | | 0.242 | 0.067 | **<0.001** | 0.110 | 0.374 | 134 | 0.046 | **<0.001** |

Notes: Relationships between validation variables and Brain-Age delta from all CU pooled subjects (including ALFA+, ADNI, EPAD and OASIS) and all MCI pooled subjects (including ADNI and EPAD). Results given by the linear model: brain-age delta ~each variable +chronological age+sex. The regression coefficients (β), standard errors (SE), P-value, 95% Confidence Interval, number of individuals (N) and effect size are depicted for each variable.

Significant values after FDR correction (*P*<0.05) are marked in bold.

Effect size in categorical variables was calculated as Cohen's D, while Cohens f² was calculated for continuous measurements. Amyloid-β status was defined by CSF (ALFA+, ADNI, and EPAD) or amyloid PET (OASIS). MCI individuals only contained individuals from ADNI and EPAD.

APOE, apolipoprotein E; WMH, White Matter Hyperintensities; CSF, cerebrospinal fluid; NfL, neurofilament light; ref, reference.

[*]Contains data from ALFA+, ADNI and EPAD.

[†]Contains data from ALFA +and ADNI.

[‡]Contains data from ADNI.

effect of sex and CSF NfL on CU brain-age delta (*Figure 4a*) did not reach significance (CU: $P_{interaction\_FDR}$=0.417 and MCI: $P_{interaction\_FDR}$=0.393).

The associations between brain-age delta and CSF NfL in CU females did not survive multiple comparisons correction (*β*=0.129, p=0.046, $P_{FDR}$ = 0.102), and was also not significant in CU males (*β*=0.006, $P_{FDR}$ = 0.999). There was not an interaction effect of sex and CSF NfL on CU brain-age delta (*Figure 4a*) ($P_{interaction\_FDR}$=0.425). In the same line, the associations between brain-age delta and longitudinal aging signature in CU females and CU males were not significant (females: $P_{FDR}$ = 0.613 and males: $P_{FDR}$ = 0.417). The interaction effect of sex on this longitudinal biomarker was not significant ($P_{interaction\_FDR}$=0.924).

**Table 5.** Relationships between validation variables and brain-age delta stratified by sex for all CU and MCI individuals.

| Model | | Females | | | | | | Males | | | | | |
|---|---|---|---|---|---|---|---|---|---|---|---|---|---|
| | | β (SE) | P-Value | [0.025, 0.075] | N | Effect size | FDR corr P-Value corr | β | P-Value | [0.025, 0.075] | N | Effect size | FDR corr P-Value Value |
| | | | | | **CU Individuals** | | | | | | | | |
| Amyloid-β pathology (ref: A-) | | 0.286 (0.065) | <0.001 | [0.160, 0.413] | 966 | 0.288 | <0.001 | 0.341 (0.076) | <0.001 | [0.192, 0.490] | 668 | 0.344 | <0.001 |
| | A+T- | 0.250 (0.082) | 0.003 | [0.088, 0.411] | | 0.252 | 0.008 | 0.2978 (0.096) | 0.002 | [0.110, 0.486] | | 0.301 | 0.006 |
| Amyloid-β / Tau pathology (ref: A-T-) | A+T + | 0.391 (0.131) | 0.003 | [0.134, 0.648] | 688 | 0.385 | 0.008 | 0.173 (0.163) | 0.290 | [-0.148, 0.494] | 474 | 0.175 | 0.424 |
| | APOE-ε2 | -0.175 (0.110) | 0.112 | [-0.391, 0.041] | | 0.172 | 0.209 | -0.061 (0.122) | 0.615 | [-0.301, 0.178] | | 0.062 | 0.766 |
| | APOE-ε4 | 0.117 (0.067) | 0.081 | [-0.014, 0.248] | | 0.116 | 0.162 | 0.249 (0.082) | 0.003 | [0.087, 0.410] | | 0.249 | 0.008 |
| APOE status (ref: APOE-ε33) | APOE-ε24 | -0.011 (0.197) | 0.958 | [-0.337, 0.358] | 966 | 0.011 | 0.999 | 0.044 (0.212) | 0.836 | [-0.372, 0.461] | 668 | 0.045 | 0.976 |
| WMH volume* | | 0.263 (0.041) | <0.001 | [0.183, 0.342] | 580 | 0.072 | <0.001 | 0.179 (0.050) | <0.001 | [0.790, 0.278] | 392 | 0.032 | <0.001 |
| CSF NfL¥ | | 0.129 (0.064) | 0.046 | [0.002, 0.256] | 228 | 0.019 | 0.102 | 0.006 (0.078) | 0.939 | [-0.149, 0.161] | 150 | 0.000 | 0.999 |
| Plasma NfL† | | 0.191 (0.059) | 0.001 | [0.076, 0.306] | 298 | 0.037 | 0.003 | 0.110 (0.069) | 0.119 | [-0.029, 0.248] | 210 | 0.012 | 0.217 |
| Brain Atrophy† | | 0.045 (0.063) | 0.475 | [-0.079, 0.169] | 171 | 0.002 | 0.635 | 0.082 (0.074) | 0.268 | [-0.064, 0.228] | 102 | 0.007 | 0.418 |
| | | | | | **MCI Individuals** | | | | | | | | |
| Amyloid-β pathology | | 0.713 (0.131) | <0.001 | [0.455, 0.970] | 217 | 0.751 | <0.001 | 0.581 (0.124) | <0.001 | [0.337, 0.824] | 286 | 0.597 | <0.001 |

*Table 5 continued on next page*

*Table 5 continued*

| | Females | | | | | | Males | | | | | |
|---|---|---|---|---|---|---|---|---|---|---|---|---|
| | β (SE) | P-value | 95% CI | N | | P-value | β (SE) | P-value | 95% CI | N | | P-value |
| **Amyloid-β / Tau pathology (ref: A-T-)** | | | | | | | | | | | | |
| A+T- | 0.556 (0.176) | 0.002 | [0.210, 0.903] | | 0.590 | 0.006 | 0.543 (0.144) | <0.001 | [0.260, 0.827] | | 0.558 | <0.001 |
| A+T + | 0.809 (0.146) | <0.001 | [0.52, 1.098] | 214 | 0.812 | <0.001 | 0.626 (0.145) | <0.001 | [0.342, 0.911] | 284 | 0.633 | <0.001 |
| APOE-ε2 | 0.034 (0.295) | 0.908 | [-0.546, 0.614] | | 0.033 | 0.999 | -0.008 (0.206) | 0.992 | [-0.408, 0.404] | | 0.002 | 0.999 |
| APOE-ε4 | 0.254 (0.142) | 0.0676 | [-0.026, 0.535] | | 0.267 | 0.155 | 0.287 (0.125) | 0.023 | [0.040, 0.534] | | 0.289 | 0.052 |
| **APOE status (ref: APOE-ε33)** | | | | | | | | | | | | |
| APOE-ε24 | 0.290 (0.418) | 0.489 | [-0.535, 1.115] | 217 | 0.309 | 0.649 | 0.399 (0.507) | 0.431 | [-0.598, 1.398] | 286 | 0.389 | 0.584 |
| WMH volume | 0.158 (0.085) | 0.063 | [-0.009, 0.325] | 181 | 0.020 | 0.133 | 0.300 (0.069) | <0.001 | [0.164, 0.437] | 252 | 0.075 | <0.001 |
| Plasma NfL‡ | 0.342 (0.098) | 0.001 | [0.147, 0.536] | 128 | 0.097 | 0.003 | 0.164 (0.091) | 0.068 | [-0.012, 0.341] | 173 | 0.020 | 0.137 |

Notes: Relationships between validation variables and Brain-Age delta from all CU pooled subjects (including ALFA+, ADNI, EPAD and OASIS) and all MCI pooled subjects (including ADNI and EPAD). Results given by the linear model: brain-age delta ~each variable +chronological age+sex. The standardized regression coefficients (β), standard errors (SE), P-value, 95% Confidence Interval, number of individuals (N) and effect size are depicted for each variable.

Significant values after FDR correction (p<0.05) are marked in bold. Effect size in categorical variables was calculated as Cohen's D, while Cohen's f² was calculated for continuous measurements. Amyloid-β status was defined by CSF (ALFA+, ADNI, and EPAD) or amyloid PET (OASIS).

APOE, apolipoprotein E; WMH, White Matter Hyperintensities; CSF, cerebrospinal fluid; NfL, neurofilament light; ref, reference.

*Contains data from ALFA+, ADNI and EPAD.

†Contains data from ALFA +and ADNI.

‡Contains data from ADNI.

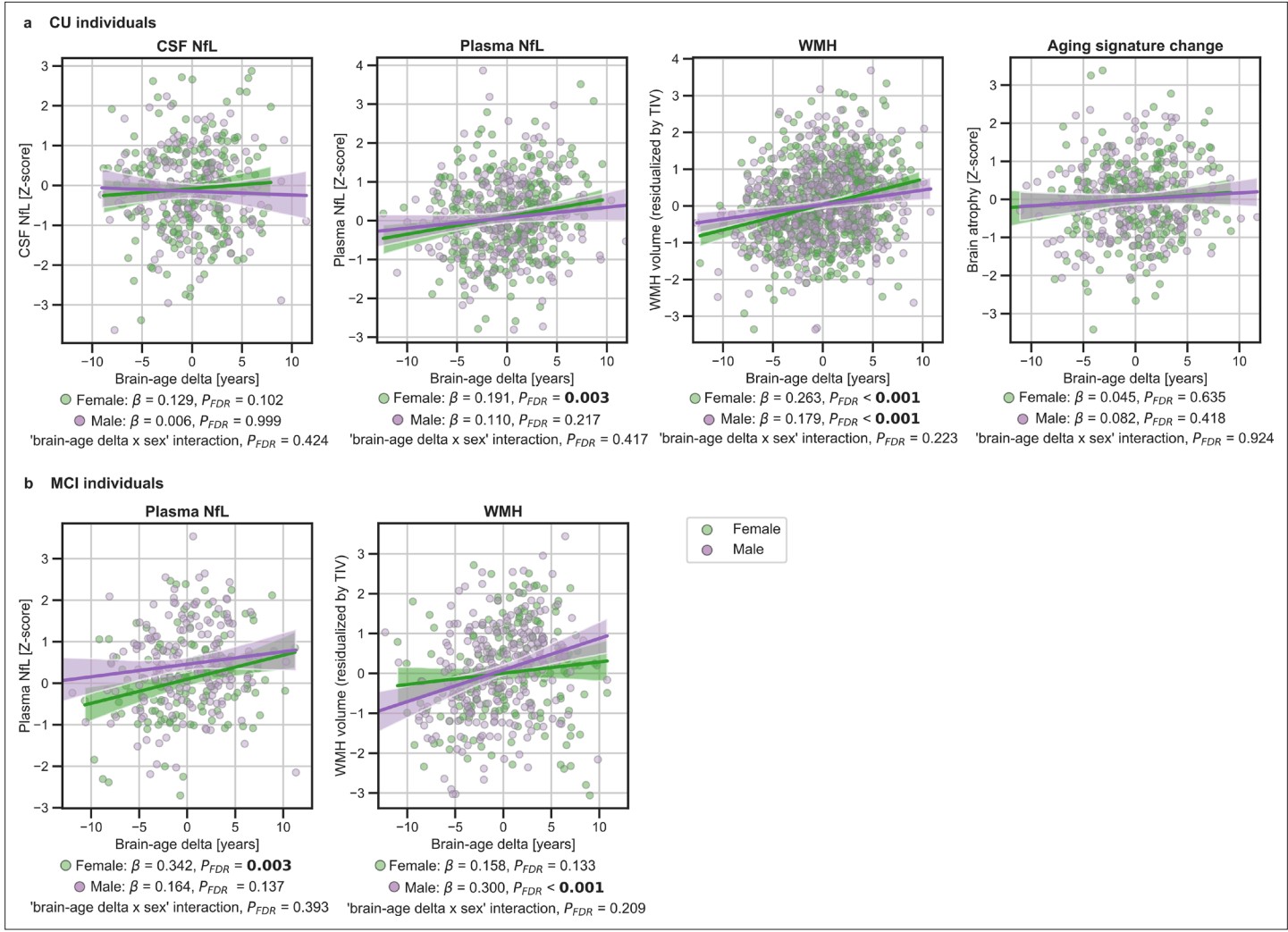

**Figure 4.** In (**a**) and (**b**), the associations of brain-age delta and continuous validation variables stratified by sex for (**a**) CU individuals and (**b**) MCI individuals. Scatter plots representing the associations of CSF NfL, plasma NfL, brain atrophy and WMH with brain-age delta in females (green) and males (purple). Each point depicts the value of the validation biomarkers of an individual and the solid lines indicate the regression line for each of the groups. 95% Confidence intervals are shown in the shaded areas. The standardized regression coefficients (β) and the corrected p-values are shown, which were computed using a linear model adjusting for age and sex. Additionally, we also computed the 'brain-age delta x sex' interaction term. The sample size for each variable can be seen in **Table 5**.

In addition, we tested the interaction between age and brain age delta to predict these biomarkers, and further stratified these analyses by sex. We found a significant interaction effect of age and CU brain-age delta on CSF NfL ($P_{\text{interaction\_FDR}}$<0.001) within the CU individuals (**Figure 5a**), by which the measures of CSF NfL were higher with age and with larger brain-age deltas (older-appearing brain). When stratifying by sex, this interaction effect of age was seen in females ($P_{\text{interaction\_FDR}}$<0.001), but not in males ($P_{\text{interaction\_FDR}}$=0.393). Regarding plasma NfL (**Figure 5b**), although we found a similar direction in the associations by which the measures of plasma NfL were higher with age and with larger brain-age deltas for CU and MCI individuals, the interaction effects were not significant ($P_{\text{interaction\_FDR}}$=0.118 and MCI: $P_{\text{interaction\_FDR}}$=0.421). When stratifying by sex, this interaction effect of age on plasma NfL was seen in CU females ($P_{\text{interaction\_FDR}}$=0.040) and not in CU males ($P_{\text{interaction\_FDR}}$=0.999). On the contrary, the interaction effect of age and plasma NfL on brain-age delta did not survive multiple comparisons in MCI males ($P_{\text{interaction}}$ = 0.022, $P_{\text{interaction\_FDR}}$=0.050). This interaction effect was also not found in MCI females ($P_{\text{interaction}}$ = 0.605, $P_{\text{interaction\_FDR}}$=0.761). The rest of the associations with neurodegeneration biomarkers not reported here were non-significant.

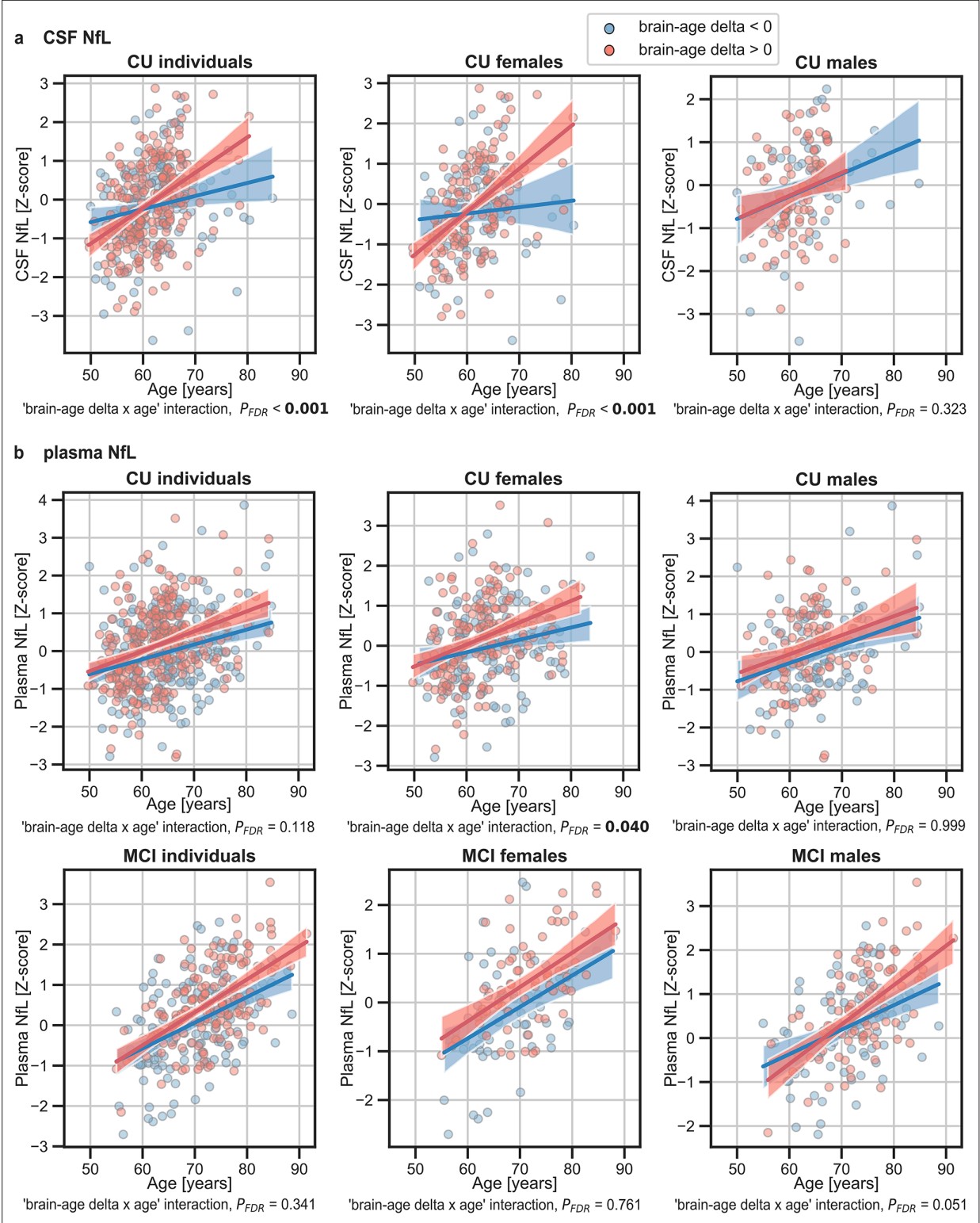

**Figure 5.** The associations of brain-age delta and (**a**) CSF NfL and (**b**) plasma NfL with chronological age for all CU and, when available, MCI individuals. For visualization purposes, individuals were categorized into two groups according to their brain-age delta: 'brain-age delta <0' representing decelerated brain aging (blue); and 'brain-age delta >0' representing accelerated brain aging (red). Scatter plots representing the associations of CSF NfL, plasma NfL and WMH with age in individuals with brain-age delta >0 and brain-age delta <0. Each point depicts the value of the validation biomarkers of an individual and the solid lines indicate the regression line for each of the groups. The regression coefficients (β) and the FDR corrected

*Figure 5 continued on next page*

*Figure 5 continued*

p-values are shown, which were computed using a linear model adjusting for age and sex. Additionally, we also computed the 'brain-age delta x sex' interaction term. The sample size for each variable can be seen in *Table 5*.

The online version of this article includes the following figure supplement(s) for figure 5:

**Figure supplement 1.** The associations of (**a**) Aging signature change and (**b**) WMH with chronological age for all CU and, when available, MCI individuals.

### Associations with markers of cerebrovascular disease

We lastly tested the associations between brain-age delta and markers of cerebrovascular disease WMH; WMH data were available in ALFA+, ADNI and EPAD. In both CU and MCI, brain-age delta was significantly associated with WMH (CU: $\beta$=0.160, $P_{FDR}$ <0.001 and MCI: $\beta$=0.222, $P_{FDR}$ <0.001) (see *Table 4*). These results were consistent with the results from the within-cohort analyses (see *Appendix 1—table 11*).

When studying the association between brain-age delta and WMH stratified by sex (*Table 5*) we found that the brain-age delta was positively associated with WMH both in CU females ($\beta$=0.263, $P_{FDR}$ <0.001) and CU males ($\beta$=0.179, $P_{FDR}$ <0.001). The interaction effect of sex and WMH on CU brain-age delta (*Figure 4a*) was not significant ($P_{interaction\_FDR}$=0.223). Conversely, we found that the brain-age delta was positively associated with WMH MCI males ($\beta$=0.300, $P_{FDR}$ <0.001), but not in females ($\beta$=0.158, $P_{FDR}$ = 0.133). The interaction effect of sex and WMH on MCI brain-age delta (*Figure 4b*) was also not significant ($P_{interaction\_FDR}$=0.209). The rest of the associations tested with WMH not reported here were non-significant.

### Discussion

In this study, we show that, in non-demented individuals, the predicted brain-age delta is associated with specific AD biomarkers (amyloid-$\beta$ PET, CSF A$\beta$42 and CSF pTau) and risk factors (*APOE* genotype), as well as with unspecific neurodegeneration biomarkers (plasma NfL), and markers of cerebrovascular disease (WMH volume). However, our results do not show that variation in cross-sectional brain-age delta was able to capture longitudinal atrophy and, therefore, it is solely a cross-sectional analysis. Our results also indicate that there might be sex differences in the development of brain aging trajectories, which must be further characterized. Taken together, our findings validate the use of machine learning predicted brain-age delta as a non-invasive biomarker of brain aging, which is associated to AD pathology in non-demented individuals with abnormal levels of biomarkers of AD and axonal injury. The capacity of brain-age delta to de detect accelerated aging must be further studied.

We have studied the associations between brain-age delta and different biomarkers of AD pathology and neurodegeneration in CU individuals. We are aware of the complexity of disentangling the effects of aging and pathology in brain aging. In this work, we do not aim to disentangle to what extent the brain structural differences are caused by AD pathology (as measured by the biomarkers that we study) or aging. Here, we show that an unspecific estimation of biological brain aging, agnostic of the underlying mechanisms is associated with the specific biological process of AD.

Regarding the associations with AD biomarkers and risk factors, regression analyses revealed significant positive associations of brain-age delta with increased A$\beta$ pathology and with AT stages for CU and MCI individuals. We also found significant associations with *APOE* status in the CU and MCI individuals, in which larger brain-age deltas were associated with the presence of *APOE*-ε4 allele. This result is in line with previous literature that has shown that *APOE*-ε4 carriership may accelerate AD-related brain atrophy (*Evans et al., 2014*; *Filippini et al., 2011*), as accelerated brain aging has also been found in MCI and AD patients (*Beheshti et al., 2018*; *Kaufmann et al., 2019*). The association of brain-age delta with *APOE*-ε4 was also previously studied, for which significant associations were found in MCI individuals (*Cole et al., 2018*; *Löwe et al., 2016*). Taken together, our results advocate for an effect of *APOE*-ε4 in physiological brain aging, albeit of a lesser magnitude than when AD pathology is present. These results with AD biomarkers and risk factors were highly reproducible in within-cohort analyses.

With the aim of studying the associations between brain-age delta and neurodegeneration, we computed the associations with NfL, a marker of neuro-axonal damage (*Khalil et al., 2018*) which can be measured both in CSF and in plasma, and with longitudinal changes in the aging signature composite as marker of age-related brain atrophy. The particular use of NfL in this context is supported by its correlation with age throughout the lifespan, as well as its strong association with all-cause mortality in the elderly (*Kaeser et al., 2021*). We found significant positive associations between brain-age delta and plasma NfL both in CU and MCI individuals, but we did not find significant associations between CSF NfL and CU brain-age delta. Even though we found NfL to be positively associated with chronological age in CU individuals, in line with previous studies (*Beheshti et al., 2018*; *Kaufmann et al., 2019*; *Khalil et al., 2020*; *Milà-Alomà et al., 2020*), the expected annual change of NfL in CU individuals whose mean age range was 65 years old is around 3.5% (*Khalil et al., 2020*). Therefore, we expected to find weak associations with brain-age delta. Still, we found a significant interaction effect of age and CSF NfL on CU brain-age delta, for which the individuals with larger brain-age delta had increased CSF NfL over the years, whereas the decelerated ones remained more stable. Taken together, a strong association between brain-age delta and plasma NfL was observed across all individuals whereas the association on CU brain-age delta and CSF NfL was milder and could only be detected as an interaction with age. Overall, these mild associations between brain-age delta and NfL suggest that the morphological effects of aging in the brain are not fully driven by neurodegeneration. In this regard, it is worth noting that cortical thinning with age has also been linked to loss of volume of the neuropil and other non-neuronal processes which are not necessarily implicated in neurodegeneration (*Vidal-Pineiro et al., 2020*). Conversely, brain-age delta was not associated with longitudinal change in aging signature. This might indicate, as previously shown (*Vidal-Pineiro et al., 2020*), that brain-age delta might not reflect accelerated brain aging but instead early life factors. However, the lack of this association might also be influenced by the short amount of time between time visits (average of 3 years) or the measurement used to measure longitudinal atrophy (cortical aging signature). Therefore, further research is needed to determine the relative contribution of early life factors and accelerated aging to brain-age delta estimates.

We also studied the associations between brain-age delta and cerebrovascular disease. Regression analyses revealed significant associations of brain-age delta with increased WMH for both CU and MCI individuals. These results were expected, as the increase in WMH with age has been previously studied (*Maniega et al., 2015*) and it has been shown that individuals with high WMH burden display spatial patterns of atrophy that partially overlap with those of brain aging (*Brugulat-Serrat et al., 2020a*; *Habes et al., 2016*). In addition, WMH have been linked to cognitive dysfunction and dementia (*Brugulat-Serrat et al., 2020b*; *Brugulat-Serrat et al., 2020a*; *Maniega et al., 2015*) and a potential pathway has been proposed, in which small vessel cerebrovascular disease affects cognition by promoting neurodegenerative changes (*Rizvi et al., 2018*). In summary, our results support an effect of cerebrovascular disease in physiological brain aging.

Brain structure aging-associated changes have been widely studied (*Bakkour et al., 2013*; *Fjell et al., 2014*). In our study, the brain regions that had highest impact on the machine learning prediction were similar to regions previously mentioned in literature (*Arenaza-Urquijo et al., 2019*; *Bakkour et al., 2013*). We found an overlap between some of our selected regions and regions included in the aging signature for both females and males, such as the precentral sulcus, insula, superior frontal and rostral middle frontal regions. In addition, the effect of sex on age-related changes in brain structure has also been studied in the recent years, with some studies reporting age–sex interactions in volumes of certain brain structures (*Coffey et al., 1998*; *DeCarli et al., 2005*), and others not finding such interactions (*Greenberg et al., 2008*). In our study, we found that, even though most of the regions with highest impact were the same for males and females, there were some regions that were sex specific. In particular, we found reduction in the superior-frontal, isthmus-cingulate and pars orbitalis regions within males and regions such as inferior-parietal, pars triangularis and paracentral within females. Most of these sex-specific regions were in concordance with previous studies (*Armstrong et al., 2019*; *Podgórski et al., 2021*). The mechanisms underlying these sex-specific brain aging differences are not well-known. Sexual hormones such as estrogen, progesterone and androgen could play a role in brain atrophy *Armstrong et al., 2019*; in the WHIMS-MRI study (*Resnick et al., 2009*), women under menopausal hormone therapy were associated with greater brain atrophy. Others, however, have proposed that estrogen and progesterone may play a protective effect in women

(*Armstrong et al., 2019*; *Green and Simpkins, 2000*). Other possible biological mechanisms influencing these results could be developmental (*Baron-Cohen et al., 2005*) or the influence of a greater presence of adverse lifestyle-related factors in men (*DeCarli et al., 2005*).

In line with the effect of sex on age-related changes in brain structure, we studied the effect of sex on the associations between brain-age delta and the above-mentioned variables. Regarding the AD biomarkers and risk factors, we found that the association between brain-age delta and a larger proportion of A+T + was only seen in females for the CU individuals, but the interaction effect of sex and AT stages on brain-age delta was not significant. Regarding the neurodegeneration variables, we found a positive association between brain-age delta and plasma NfL in CU and in MCI females. These associations were not seen in their male counterparts. However, the interaction effect of sex and CSF and plasma NfL on CU and MCI brain-age delta was not significant. In addition, we found a significant interaction effect of age and CSF and plasma NfL on brain-age delta, for which the CU females with larger brain-age delta had increased CSF and plasma NfL over the years, whereas the decelerated ones remained more stable. These results were expected, as females have higher chances of undergoing neurodegeneration and have showed to undergo faster cognitive decline than males (*Ferretti et al., 2018*). Although the role of sex hormones still needs to be clarified, it has been suggested that the menopausal drop of estrogen increases vulnerability to neurological events (*Green and Simpkins, 2000*; *Maioli et al., 2021*). On the contrary, results suggest that morphological effects of aging in the CU males' brain are not fully driven by neurodegeneration. Lastly, regarding the cerebrovascular disease biomarkers, we found a positive association between brain-age delta and WMH for both CU females and males, while no interaction effect of sex was found. Conversely, in MCI individuals, we only found positive associations between brain-age delta and WMH on males, but no interaction effect of sex was found in MCI individuals. Overall, we found cross-sectional sex differences in the associations between brain-age delta and markers of neurodegeneration and cerebrovascular disease. NfL was only positively associated with brain-age delta in females, while WMH were positively associated with both CU females and males. Positive associations between NfL and WMH have been previously demonstrated, for both CU and MCI (*Osborn et al., 2018*), and the different AD stages (*Walsh et al., 2021*). Moreover, it has been proposed that WMH may reflect two different pathological pathways, one including amyloid aggregation and another including axonal injury (*Osborn et al., 2018*). Our results may suggest that brain aging in males might be driven more strongly by the former pathway, while brain aging in females might be driven more by the latter one. However, the lack of significant brain-age delta-by-sex interactions in our analyses reflect the limitations in the evidence for the above-mentioned sex-differences. Further work exploring these sex differences in detail is warranted.

Our purpose was to study the clinical validity of using the brain-age delta as a proxy biomarker of brain aging associated to AD and neurodegeneration. Therefore, our main aim was studying the characteristics of the individuals whose brain age is more accelerated or decelerated. One of the strengths of this study was the robustness of the brain-age delta measurement using a widely used segmentation atlas such as the Desikan-Killiany. Notably, we demonstrated the robustness of our method by training our model with one cohort and testing independently on four independent cohorts. The similar results obtained in all cohorts allowed us to seek associations in a large sample of participants with biomarker data and to further stratify the data by sex. This aspect is critical for this type of analyses as the effects of biological aging are necessarily very small, particularly in CU individuals of limited age range. This may explain why we could not detect significant effects versus longitudinal brain atrophy, as the available sample size for these analyses was smaller since this variable was not available in all cohorts. Another strength of our work was that we were able to include a wide range of different biomarkers of AD pathology allowing us to perform an in-depth analysis of the effect of these measurements with the brain-age delta. Conversely, our model used a smaller number of features and a training set with a more limited age range than other models seen in literature recently (*Liem et al., 2017*; *Peng et al., 2021*), leading to a performance which cannot be compared against these state-of-the-art models. However, the performance of age prediction was similar to other publications that used similar methodologies (*Beheshti et al., 2018*; *Dafflon et al., 2020*; *de Lange et al., 2020a*) and, most importantly, was successful in studying the utility of brain-age delta as biomarker of brain aging in non-demented individuals, which is impacted by abnormal levels of biomarkers of AD and neurodegeneration. Moreover, using a relatively restricted number of features allowed us to visualize and interpret in an easier way the contribution of different brain regions to the brain age

prediction by means of SHAP values. This ensured that the model had neurobiological interpretability. However, it is important to note that SHAP values may not reflect the true relationship between the features and the model's prediction in the presence of correlated features. This is an important consideration to bear in mind for correctly interpreting our results. In addition, future work should also focus on developing a model with larger number of features or a 3D model and should study the effect of these validation measurements for AD and neurodegeneration, as well as longitudinal changes, with the brain-age delta more in depth.

In conclusion, we validated that machine-learning based brain age prediction obtained from a widely used segmentation atlas can be used as a biomarker of biological brain aging associated with AD pathology, risk factors, and neurodegeneration. Moreover, our results confirm the presence of sex-related brain aging structural changes and suggest the prevalence of different neuropathological pathways involved in brain aging within females and males. Therefore, these results indicate the importance on considering different approaches for assessing aging and neurodegeneration differently for each sex.

# Materials and methods

## Participants

We used a collection of T1-weighted brain MRI scans included in the UK Biobank (https://www.ukbiobank.ac.uk) cohort for training the proposed model and for calculating cross-validated brain age predictions. The dataset consisted of CU individuals ($N$ = 22,661), after excluding subjects with ICD-9 and ICD-10 diagnosis, covering individuals of ages 44–81.

We also used four different cohorts to investigate the association between brain-age deltas with different sets of biomarker and AD risk factor measurements. Inclusion criteria for the independent cohorts consisted of: (i) availability of T1-weighted MRI brain scans; (ii) and availability of apolipoprotein E (*APOE*) categories and of CSF or PET measures for amyloid-$\beta$ pathology acquired in less than a year from the MRI acquisition. These cohorts included individuals with different diagnosis: CU and MCI. ADNI cohort included CU and MCI individuals from ADNI 1,2 and 3 (N=751, CU = 253, MCI = 498), the EPAD cohort included CU and MCI individuals (N=808, CU = 653, MCI = 155), the ALFA + cohort included only CU individuals (N=380) and the OASIS cohort included CU individuals (N=407). MCI individuals, which were only included from ADNI and EPAD cohorts, were specified by a Clinical Dementia Rating = 0.5. We did not include individuals with a dementia diagnosis in the AD continuum because we wanted to focus on assessing the impact of abnormal AD biomarkers and risk factors on brain-age estimates in preclinical and prodromal AD stages.

All the individuals had available data for the following clinical variables: chronological age, sex, MMSE and years of education, which will be referred as clinical variables from now on. A more detailed description of the clinical variables of these datasets is given in *Table 1*. Regarding AD-related variables, ALFA+, ADNI and EPAD cohorts included CSF Aβ42 measurements for categorizing Aβ pathology status, AT status determined by CSF Aβ42 and CSF p-tau, *APOE* categories and WMH. OASIS, meanwhile, only had data available for Aβ PET and *APOE* categories. In addition, ALFA + and ADNI included biomarkers of neurodegeneration such as CSF NfL, plasma NfL and cortical atrophy measured by longitudinal changes in the so-called aging signature (*Bakkour et al., 2013*). The combination of available AD-related variables and neurodegeneration biomarkers will be referred as validation variables from so on. A more detailed description of the validation variables can be seen in *Table 2*.

## Image acquisition and preprocessing

The UK Biobank, ADNI and OASIS datasets had available T1-weighted magnetic resonance (MR) images that had already been segmented with Freesurfer and had been parcellated using the FreeSurfer's cortical Desikan-Killiany (*Desikan et al., 2006*) and subcortical aseg (*Fischl et al., 2002*) labeling pipelines, which had undergone a quality control procedure. Taking advantage of this available data, we decided to use the same segmentation pipeline with the ALFA + and EPAD cohorts. All the image acquisition and preprocessing done is as follows.

The UK Biobank dataset consisted of T1-weighted magnetic resonance (MR) images, all collected using a 3T Siemens Skyra scanner and preprocessed as previously explained in more detail (https://

biobank.ctsu.ox.ac.uk/crystal/crystal/docs/brain_mri.pdf). Images were previously segmented with Freesurfer 6.0 and underwent a quality control procedure.

For ADNI participants (*Petersen et al., 2010*), MRI acquisition methods are described in more detail elsewhere (http://adni.loni.usc.edu/methods/documents/). In brief, most of the T1-weighted MR were MP-RAGE, acquired with 1.5T or 3T scanners. Images were segmented with Freesurfer 5.1 and 6.0 and subjected to a quality control procedure. When possible, we also included a second T1-weighted MRI image sequence for the participants that underwent another MRI visit 3 years later. These scans were also segmented following the previously explained procedure.

For the OASIS subjects (*Marcus et al., 2007*), the MRI scans were acquired on 1.5T or on 3.0T scanners. T1-weighted magnetization-prepared rapid gradient echo (MP-RAGE) scans were obtained according to previously explained protocol (https://theunitedconsortium.com/wp-content/uploads/2021/07/OASIS-3_Imaging_Data_Dictionary_v1.8.pdf). All MRI sessions were segmented using Free-Surfer 5.1 or 5.3 and followed quality control measures. The PET images were acquired with [11 C] PIB Pittsburgh's compound 60-min dynamic PET scan in 3D mode and the corresponding analysis analyses were performed using the PET unified pipeline (PUP, https://github.com/ysu001/PUP) (*Su, 2018*). Mean standardized uptake value (SUVR) values were converted to Centiloid scale as previously explained, using the whole cerebellum as reference region.

For the ALFA + participants, a high-resolution 3D T1-weighted MRI sequence was acquired in a 3T Philips Ingenia CX scanner (TE/TR = 4.6/9.9ms, Flip Angle = 8°; voxel size = 0.75 × 0.75 x0.75 mm). Images were segmented with Freesurfer 6.0 and subjected to a quality control procedure to identify and remove incidental findings (*Brugulat-Serrat et al., 2017*) and segmentation errors (*Huguet et al., 2021*). Some of these ALFA + subjects (N=187) underwent a second MRI visit 3 years after the initial visit, where another T1-weighted MRI sequence was acquired and segmented following the same procedure as in the first visit.

For the EPAD cohort (*Solomon et al., 2018*), which is a multisite study, T1-weighted MRIs were inversion-recovery prepare 3D gradient-echo sequences, acquired with 3T scanners. Images were segmented with Freesurfer 6.0 and subjected to a quality control procedure (*Lorenzini et al., 2021*).

For all the cohorts, subsequent to the FreeSurfer segmentation, tissue regions were parcellated into 183 different anatomical regions of interest (ROI)s using the widely-used FreeSurfer's cortical Desikan-Killiany (*Desikan et al., 2006*) and subcortical aseg (*Fischl et al., 2002*) labeling pipelines. As mentioned before, we used the available FreeSurfer segmentations from UK Biobank, ADNI, and OASIS cohorts. All volumes were residualized with respect to total intracranial volume (TIV) and to scanning site, while all cortical thicknesses were residualized with respect to scanning site, using linear models. Lastly, we performed a standardization procedure by computing z-score measurements feature-wise within each cohort, as previously performed (*Casamitjana et al., 2018*; *Subramaniapillai et al., 2021*; *Ten Kate et al., 2018*). We then assessed that there were not statistical differences in mean cortical thickness and volumes between the cohorts (see *Appendix 1—figure 1*).

## Biomarkers
### CSF and plasma collection, processing and biomarkers measurements
CSF and blood collection, processing and storage in the ALFA + study have been described previously (*Milà-Alomà et al., 2020*; *Suárez-Calvet et al., 2020*). CSF p-tau181 was measured using the Elecsys Phospho-Tau (181 P) CSF electrochemiluminescence immunoassay on a fully automated cobas e 601 instrument (Roche Diagnostics International Ltd, Rotkreuz, Switzerland). CSF Aβ42 and NfL were measured with the NeuroToolKit on a cobas e 411 or cobas e 601 instrument (Roche Diagnostics International Ltd, Rotkreuz, Switzerland). Plasma NfL was measured using the commercial Quanterix assay (Simoa NF-light Kit cat. no. 103186) on a HD-X analyzer following the manufacturer's instructions (Quanterix, Billerica, MA, USA). All these measurements were previously reported (*Milà-Alomà et al., 2020*; *Suárez-Calvet et al., 2020*). All measurements were performed at the Clinical Neurochemistry Laboratory, University of Gothenburg, Mölndal, Sweden, by laboratory technicians and scientists blinded to participants' clinical information.

In the ADNI study, CSF samples were measured according to the kit manufacturer's instructions and as described in previous studies (*Bittner et al., 2016*), using the Elecsys β-amyloid(1–42) CSF (*Bittner et al., 2016*), and the Elecsys Phospho-Tau (181 P) and Elecsys Total-Tau CSF immunoassays on a cobas e 601 analyzer at the Biomarker Research Laboratory, University of Pennsylvania, USA. Plasma

NfL was measured on an in-house immunoassay on the single-molecule array (Simoa) platform, using the same methodology as described previously, at the Clinical Neurochemistry Laboratory, University of Gothenburg, Mölndal, Sweden.

In the EPAD study, CSF was measured using the Elecsys β-amyloid (1–42) and the Elecsys Phospho-Tau (181 P) CSF electrochemiluminescence immunoassay on a fully automated cobas e 601 instrument (Roche Diagnostics International Ltd.). All measurements were performed at the Clinical Neurochemistry Laboratory, University of Gothenburg, Mölndal, Sweden, by laboratory technicians and scientists blinded to participants' clinical information. Concentrations of CSF Aβ42 and p-tau181 were determined according to the manufacturer's instructions (**Solomon et al., 2018**).

## Amyloid-β positivity cutoffs

For ALFA+, ADNI and EPAD, AT stages were defined by CSF Aβ42 and CSF p-tau, respectively. Previously used cut-offs were applied to each cohort, consisting of 1098 pg/mL for CSF Aβ42 for ALFA + and EPAD (**Schindler et al., 2018**) and of 880 pg/mL for CSF Aβ42 for ADNI (**Hansson et al., 2018**) and of 24 pg/mL for p-tau for the three cohorts (**Milà-Alomà et al., 2020**). For OASIS, we used the cut-off value of 17 Centiloids from literature (**Salvadó et al., 2019**).

## WMH volumes

WMH volumes were generated for ALFA + and EPAD cohorts using Bayesian Model Selection (BaMoS) procedure (**Sudre et al., 2015**), which has been provided previously. We also obtained the already available WMH volumes for ADNI cohort, in which the method of WMH volumetric quantification was performed using probabilistic models in a Markov Random Field framework, as previously provided (**Schwarz et al., 2009**). For each cohort, total WMH volumes were derived by summing and multiplying the number of labeled voxels by voxel dimensions. Total WMH volumes were natural log transformed and residualized with respect to TIV using linear models.

## Aging signature measurements

For ALFA + and ADNI, we computed the weighted Dickerson's aging signature (**Bakkour et al., 2013**), which has been used as a proxy measurement for brain aging. The aging signature is a map of specific brain regions that undergo cortical atrophy in normal aging. This meta-ROI is composed of the surface-area weighted average of the mean cortical thickness in the following individual ROIs: calcarine, caudal fusiform, caudal insula, cuneus, inferior frontal gyrus, medial superior frontal and precentral cortices. A Z-score of this aging-specific measure was calculated based on the mean and standard deviation of the CU individuals, as done previously (**Bakkour et al., 2013**). This is referred as Aging Signature V1.

In addition, we also computed this measurement on the scanners from the second MRI visit, referred as Aging Signature V2. We then computed a longitudinal brain atrophy measurement by computing the aging signature change over the years between the MRI acquisitions. Therefore, longitudinal aging signature change, or brain atrophy, was computed as:

$$\text{Aging signature change} = \frac{\text{aging signature Visit 2} - \text{aging signature Visit 1}}{\text{Time between visits}}$$

For a secondary analysis shown in *Appendix 1—table 7*, we also computed the aging signature (Aging Signature V1) for the remaining independent cohorts: EPAD and OASIS.

## Brain-age prediction

### Regression model

#### Model workflow

For the current study, we used a gradient boosting framework: the XGBoost regressor model from the XGBoost python package (https://xgboost.readthedocs.io/en/) to run the brain age prediction. This regressor, which is based on a decision-tree based ensemble algorithm, was selected due to its speed and performance and its advanced regularization to reduce overfitting (**Chen and Guestrin, 2016**). In addition, large-scale brain age studies have demonstrated its adequacy (**Bashyam et al., 2020**; **de Lange et al., 2019**; **de Lange et al., 2020b**; **Kaufmann et al., 2019**). As it has been shown that there are sex-related trajectories in normal aging (**Podgórski et al., 2021**), we trained separate

models for females and males. For each model, we first performed Bayesian parameter optimization based on a cross-validation scheme with ten folds repeated ten times using the FreeSurfer volumes and thickness of the UK Biobank as input. For the optimization we used HYPEROPT (*Bergstra et al., 2013*), with which we scanned for maximum depth, number of estimators, learning rate, alpha regularization, lambda regularization, subsample, gamma and colsample by tree. The optimized parameters were maximum depth = 4, number of estimators = 800, learning rate = 0.03, alpha regularization = 4, lambda regularization = 1, subsample = 0.36, gamma = 3 and colsample by tree = 0.89 for the males model; and maximum depth = 4, number of estimators = 850, learning rate = 0.03, alpha regularization = 8.5, lambda regularization = 14.5, subsample = 0.449, gamma = 3.5 and colsample by tree = 0.72 for the females model. We trained these two models and performed the brain-age prediction on the independent cohorts. We decided to compute a ROI based model using these 183 FreeSurfer regions because they are widely used and available in most of the neuroimaging datasets. Therefore, our aim was not to compare our performance to the one achieved by a model trained with larger number of ROIs or with the full 3D images, but to study the generalizability and the relevance of our model in the AD field.

## Contribution of brain regions in prediction

We computed SHAP (SHapley Additive exPlanation) values (https://github.com/slundberg/shap) (*Lundberg et al., 2020*) to measure the contribution of each brain region in the prediction of age for each subject. SHAP assigns an importance value within the prediction to each feature (in this case, brain region), which is based on its unique consistent and locally accurate attribution (*Lundberg et al., 2020*). We calculated the average SHAP value for each region for all females and males of the UK Biobank cohort.

In addition, to assess that the regions with highest SHAP values were stable, we performed a permutation approach to study the significance of each region, separately for females and males. With this aim we compared the averaged SHAP value (region-specific) obtained when using the entire train set on the model to a null distribution calculated from 1,000 permutations performing subsample of the subjects, in which we trained and tested the model using 80% and 20% of the individuals, respectively.

## Brain-age delta estimation

We predicted brain-age on the independent cohorts separately: ALFA+, EPAD, ADNI, and OASIS, using the previously trained model. To investigate the prediction performance, correlation analyses were run for predicted brain-age versus chronological age, $R^2$, root mean square error (RMSE), and mean absolute error (MAE) were calculated for each independent cohort separately, as well as for females and males separate pooled from all independent cohorts. We also investigated the prediction performance on the UK Biobank cohort by computing the average latter metrics from a cross-validation with 10 splits and 10 repetitions.

As recent research has shown that brain-age estimation involves a proportional bias (*de Lange et al., 2020a*; *Le et al., 2018*; *Liang et al., 2019*; *Smith et al., 2019*), we applied a well-established age-bias correction procedure to our data (*de Lange et al., 2020a*; *Le et al., 2018*). This correction, as originally proposed (*de Lange and Cole, 2020c*; *Le et al., 2018*), consists of a linear regression between age ($\Omega$) and brain-predicted age ($Y$) on each of the independent cohorts, $Y = \alpha \times \Omega + \beta$. The derived values of slope (α) and intercept (β) from the training set were used to correct the predicted brain-age in each test set by applying: $\text{Corrected Predicted Brain Age} = \text{Predicted Brain Age} + [\Omega - (\alpha \times \Omega + \beta)]$. By subtracting the chronological age from the Corrected Predicted Brain Age, we obtained the brain-age delta which was used to test the associations with the validation measurements.

## Statistical analyses

All statistical analyses were conducted using Python 3.7.0. We tested for normality of the distribution for each biomarker using the Kolmogorov-Smirnov test and visual inspection of histograms. CSF NfL and plasma NfL did not follow a normal distribution and were thus natural log transformed. In addition, to compare the measurements for CSF NfL and for plasma NfL coming from different cohorts (ALFA +and ADNI), CSF and plasma NfL was converted to z-scores.

To study the performance and accuracy of the brain-age prediction for each cohort, correlation analyses were run for predicted brain-age versus chronological age. We also computed $R^2$, RMSE and MAE for each cohort, as well as the age bias of the prediction after bias correction. We assessed statistically whether the accuracy of the predicted brain-age was different between cohorts by using Fisher's z-transformation for correlation coefficients. In addition, we computed these performance metrics to assess the differences in the model for females and males in the pooled cohorts. Results from another secondary analysis are also shown in *Appendix 1—table 7*, in which we assessed the performance and accuracy of the aging signature for all cohorts, by performing correlation analyses between the aging signature versus chronological age and computing the $R^2$ and RMSE. In this secondary analyses, we also studied whether the performance obtained for the predicted brain-age was better than the aging signature by performing the William's test (*Williams, 1959*) for the Pearson's correlation coefficient and a F-test to assess which model was statistically better (*Appendix 1—table 7*).

We used the brain-age delta as a measure of brain aging to study the associations between this measurement and the different AD and neurodegeneration biomarkers and risk factors. With this aim, we pooled all the subjects from all cohorts together and computed linear regression models for each validation variable, in which chronological age and sex were included as covariates. Local effect size of each of the continuous validation variables was calculated using Cohen's $f^2$ (*Cohen, 2013*). The mean brain age delta among Aβ pathology, AT stages and *APOE* status, were assessed by a one-way analysis of covariance (ANCOVA) adjusting for age and sex. Effect size of the different levels was calculated by dividing the estimated difference in the brain-age delta between the different categories by the estimated standard deviation. We also stratified the individuals by sex and studied the associations between brain-age delta and the different validation variables by computing linear regression models in which chronological age was included as covariate. We next tested for interactions between sex and the validation variables on brain-age delta using linear regression models and including chronological age as covariate. Lastly, we tested for interactions between age and the validation variables on brain-age delta for CU and MCI individuals, as well as for females and males. Correction for multiple comparisons was performed using false discovery rate correction (FDR) (*Benjamini and Hochberg, 1995*). The number of tests for which we corrected was 125.

We also performed two different secondary and exploratory analyses. First, we assessed whether the brain regions (volumes and cortical thickness) that contributed the most to the prediction of females and males in the UK Biobank were different for each sex. With this aim we performed regression models to study the interaction effect of sex for each of the selected ROIs by including a sex-age interaction term.

Secondly, we wanted to identify the individuals whose predicted brain-age deviate the most from chronological aging, that is, individuals with the highest positive or lowest negative brain-age deltas, to study the above-mentioned associations. The aim of this secondary analysis was to examine the impact of these extreme individuals on the overall relationships observed in our sample, in the expectation that the observed effect would be small. Even though this procedure implies that some of the data is discarded, we expected that the resulting analyses may be more sensitive. With this aim, we selected the individuals whose brain-age delta was included within the 10th and 90th percentile of the distribution for each independent cohort and studied the differences between these groups. The methodology and the results of this analysis can be found in Appendix 1.

## Acknowledgements

This publication is part of the ALFA study (ALzheimer and FAmilies). The authors would like to express their most sincere gratitude to the ALFA project participants, without whom this research would have not been possible. Authors would like to thank Roche Diagnostics International Ltd. for kindly providing the kits for the CSF analysis of ALFA + participants and GE Healthcare for kindly providing [18 F] flutemetamol doses of ALFA + participants. Collaborators of the ALFA Study are: Müge Akinci, Annabella Beteta, Alba Cañas, Irene Cumplido, Carme Deulofeu, Ruth Dominguez, Maria Emilio, Karine Fauria, Sherezade Fuentes, Oriol Grau-Rivera, Laura Hernandez, Gema Huesa, Jordi Huguet, Eider M Arenaza-Urquijo, Eva M Palacios, Paula Marne, Tania Menchón, Carolina Minguillon, Eleni Palpatzis, Cleofé Peña-Gómez, Albina Polo, Sandra Pradas, Blanca Rodríguez-Fernández, Aleix Sala-Vila, Gemma Salvadó, Mahnaz Shekari, Anna Soteras, Laura Stankeviciute, Marc Vilanova and Natalia Vilor-Tejedor.

The project leading to these results has received funding from "la Caixa" Foundation (ID 100010434), under agreement LCF/PR/GN17/50300004 and the Alzheimer's Association and an international anonymous charity foundation through the TriBEKa Imaging Platform project (TriBEKa-17–519007). Additional support has been received from the Universities and Research Secretariat, Ministry of Business and Knowledge of the Catalan Government under the grant no. 2017-SGR-892 and the Spanish Research Agency (AEI) under project PID2020-116907RB-I00 of the call MCIN/ AEI /10.13039/501100011033. FB is supported by the NIHR biomedical research center at UCLH. MSC receives funding from the European Research Council (ERC) under the European Union's Horizon 2020 research and innovation programme (Grant agreement No. 948677), the Instituto de Salud Carlos III (PI19/00155), and from a fellowship from "la Caixa" Foundation (ID 100010434) and from the European Union's Horizon 2020 research and innovation programme under the Marie Skłodowska-Curie grant agreement No 847648 (LCF/BQ/PR21/11840004).

COBAS, COBAS E and ELECSYS are trademarks of Roche. All other product names and trademarks are the property of their respective owners. The Roche NeuroToolKit is a panel of exploratory prototype assays designed to robustly evaluate biomarkers associated with key pathologic events characteristic of AD and other neurological disorders, used for research purposes only and not approved for clinical use.

The authors would like to thank the reviewers for their constructive input on this manuscript.

# Additional information

### Competing interests

José Luis Molinuevo: is currently a full-time employee of H. Lundbeck A/S and previously has served as a consultant or on advisory boards for the following for-profit companies or has given lectures in symposia sponsored by the following for-profit companies: Roche Diagnostics, Genentech, Novartis, Lundbeck, Oryzon, Biogen, Lilly, Janssen, Green Valley, MSD, Eisai, Alector, BioCross, GE Healthcare, and ProMIS Neurosciences. Adam J Schwarz: Employee of Takeda Pharmaceutical Company Ltd. Ivonne Suridjan: Is a full-time employee and shareholder of Roche Diagnostics International Ltd. Gwendlyn Kollmorgen: Is a full-time employee of Roche Diagnostics GmbH. Anna Bayfield: Is a full-time employee and shareholder of Roche Diagnostics GmbH. Marc Suárez-Calvet: Has served as a consultant and at advisory boards for Roche Diagnostics International Ltd and has given lectures in symposia sponsored by Roche Diagnostics, S.L.U, Roche Farma, S.A and Roche Sistemas de Diagnósticos, Sociedade Unipessoal, Lda. ALFA study: EPAD study: ADNI study: OASIS study: The other authors declare that no competing interests exist.

### Funding

| Funder | Grant reference number | Author |
| --- | --- | --- |
| Horizon 2020 - Research and Innovation Framework Programme | 948677 | Marc Suárez-Calvet |
| Instituto de Salud Carlos III | PI19/00155 | Marc Suárez-Calvet |
| La Caixa Foundation | 100010434 | Marc Suárez-Calvet |
| Horizon 2020 - Research and Innovation Framework Programme | 847648 | Marc Suárez-Calvet |
| EU/EFPIA Innovative Medicines Initiative Joint Undertaking AMYPAD | 115952 | Juan Domingo Gispert |
| La Caixa Foundation | 100010434; LCF/PR/ GN17/50300004 | Marc Suárez-Calvet |
| TriBEKa Imaging Platform project | TriBEKa-17-519007 | Irene Cumplido-Mayoral |

| Funder | Grant reference number | Author |
|---|---|---|
| Universities and Research Secretariat, Ministry of Business and Knowledge of the Catalan Government | 2017-SGR-892 | Irene Cumplido-Mayoral |
| EIT Digital | Grant 2021 | Juan Domingo Gispert |
| Spanish Ministry of Science and Innovation (MCIN)/Spanish Research Agency (AEI) MCIN/AEI /10.13039/501100011033 | RTI2018-102261, co-funded by the European Regional Development Fund (FEDER) | Juan Domingo Gispert |
| NIHR biomedical research center at UCLH | | Frederik Barkhof |
| Instituto de Salud Carlos III | PI22/00456 | Marc Suárez-Calvet |
| Spanish Ministry of Science and Innovation (MCIN)/Spanish Research Agency (AEI) MCIN/ AEI/10.13039/501100011033 | PID2021-125433OA-100 | Raffaele Cacciaglia |
| Spanish Research Agency (AEI) | PID2020-116907RB-I00 of the call MCIN/ AEI /10.13039/501100011033 | Verónica Vilaplana |

The funders had no role in study design, data collection and interpretation, or the decision to submit the work for publication.

### Author contributions

Irene Cumplido-Mayoral, Conceptualization, Formal analysis, Investigation, Methodology, Writing – original draft; Marina García-Prat, Data curation, Software, Visualization, Writing – review and editing; Grégory Operto, Luigi Lorenzini, Silvia Ingala, Alle Meije Wink, Resources, Data curation, Software, Writing – review and editing; Carles Falcon, Mahnaz Shekari, Marta Milà-Alomà, Resources, Data curation, Investigation, Writing – review and editing; Raffaele Cacciaglia, Validation, Investigation, Visualization, Methodology, Writing – review and editing; Henk JMM Mutsaerts, Validation, Investigation, Visualization, Writing – review and editing; Carolina Minguillón, Karine Fauria, Resources, Data curation, Project administration, Writing – review and editing; José Luis Molinuevo, Validation, Investigation, Writing – review and editing; Sven Haller, Gael Chetelat, Adam Waldman, Adam J Schwarz, Frederik Barkhof, Resources, Validation, Investigation, Writing – review and editing; Ivonne Suridjan, Gwendlyn Kollmorgen, Anna Bayfield, Resources, Writing – review and editing; Henrik Zetterberg, Kaj Blennow, Resources, Data curation, Validation, Investigation, Writing – review and editing; Marc Suárez-Calvet, Conceptualization, Data curation, Validation, Investigation, Writing – original draft, Writing – review and editing; Verónica Vilaplana, Juan Domingo Gispert, Conceptualization, Supervision, Validation, Investigation, Methodology, Writing – original draft, Writing – review and editing; ALFA study, EPAD study, ADNI study, OASIS study, Resources;

### Author ORCIDs

Irene Cumplido-Mayoral ⓘ http://orcid.org/0000-0002-1967-7025
Henk JMM Mutsaerts ⓘ http://orcid.org/0000-0003-0894-0307
Sven Haller ⓘ http://orcid.org/0000-0001-7433-0203
Adam J Schwarz ⓘ http://orcid.org/0000-0002-9743-6171
Verónica Vilaplana ⓘ http://orcid.org/0000-0001-6924-9961
Juan Domingo Gispert ⓘ http://orcid.org/0000-0002-6155-0642

### Ethics

Clinical trial registration Clinicaltrials.gov Identifier: NCT01835717.
Human subjects: ALFA ethics: All participants were enrolled in the ALFA (ALzheimer and FAmilies) study (Clinicaltrials.gov Identifier: NCT01835717). The study was approved by the Independent Ethics Committee "Parc de Salut Mar," Barcelona, and all participants gave written informed consent.

Decision letter and Author response
Decision letter https://doi.org/10.7554/eLife.81067.sa1
Author response https://doi.org/10.7554/eLife.81067.sa2

---

## Additional files

### Supplementary files
• MDAR checklist

### Data availability
UK Biobank data availability at https://www.ukbiobank.ac.uk/ ADNI data availability at https://adni.loni.usc.edu/ EPAD data availability at https://ep-ad.org/ ALFA: data availability through GAAIN at https://www.gaaindata.org/partners/online.html Fully steps and instructions on data access can be found in the following links: UK Biobank: https://www.ukbiobank.ac.uk/enable-your-research/apply-for-access ADNI: https://adni.loni.usc.edu/data-samples/access-data/EPAD: https://ep-ad.org/open-access-data/access/ ALFA: https://www.gaaindata.org/partner/ALFA.

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

# Appendix 1

**Appendix 1—table 1.** Correlations between validation variables and chronological age.

|  | Aging signature | | Amyloid-β | | Log WMH | | Log Plasma NfL | | Log CSF NfL | |
|---|---|---|---|---|---|---|---|---|---|---|
|  | R | P-value | R | P-value | R | P-value | R | P-value | R | P-value |
| ALFA+ | –0.23 | **<0.001** | –0.12 | **0.026** | 0.25 | **<0.001** | 0.322 | **<0.001** | 0.403 | 0.352 |
| ADNI | –0.256 | **<0.001** | –0.100 | 0.123 | 0.370 | **<0.001** | 0.441 | **<0.001** | - | - |
| EPAD | –0.412 | **<0.001** | –0.04 | 0.278 | 0.401 | **<0.001** | - | - | - | - |
| OASIS | –0.455 | **<0.001** | 0.278 | **<0.001** | - | - | - | - | - | - |

The Pearson's correlation coefficient (R) between different validation variables (aging signature, amyloid-β, WMH, plasma NfL, CSF NfL) and chronological age for the independent cohorts used: ALFA+, ADNI, EPAD and OASIS. Significant values ($P<0.05$) are marked in bold.
WMH, White Matter Hyperintensities; NfL, neurofilament light, CSF, cerebrospinal fluid.

**Appendix 1—table 2.** Comparison of prediction's metrics across cohorts.

|  | Fisher's z | *P*-Value |
|---|---|---|
| ALFA +vs ADNI | –2.446 | 0.993 |
| ALFA +vs_EPAD | –5.291 | 1 |
| ALFA +vs OASIS | –8.630 | 1 |
| ADNI vs EPAD | –1.817 | 0.965 |
| ADNI vs OASIS | –4.955 | 1 |
| EPAD vs OASIS | –4.166 | 0.999 |

Testing whether the Pearson's correlation coefficients from the brain-age prediction against the chronological age is similar across all cohorts, via Fisher's transformation.

**Appendix 1—table 3.** Prediction metrics for different diagnostic groups in the different cohorts.

| Cohorts | Correlation with age | | MAE (y) | $R^2$ | RMSE |
|---|---|---|---|---|---|
|  | R | P-value |  |  |  |
| | | | CU individuals | | |
| ADNI | 0.575 | **<0.001** | 8.153 | 0.331 | 4.953 |
| EPAD | 0.634 | **<0.001** | 4.494 | 0.402 | 5.456 |
| | | | MCI individuals | | |
| ADNI | 0.599 | **<0.001** | 6.739 | 0.358 | 5.760 |
| EPAD | 0.557 | **<0.001** | 5.22 | 0.311 | 5.871 |

The Pearson's correlation coefficient (R) between predicted brain-age and chronological age, R2 , root mean square error (RMSE), and mean absolute error (MAE) for CU and MCI individuals from ADNI and EPAD.

**Appendix 1—table 4.** Prediction metrics for females and males in training set.

|  | $MAE_{orig}$ | $RMSE_{orig}$ | $R^2_{orig}$ | $MAE_{corr}$ | $RMSE_{corr}$ | $R^2_{corr}$ |
|---|---|---|---|---|---|---|
| Female | 4.221 (0.059) | 5.291 (0.062) | 0.696 (0.006) | 2.871 (0.062) | 3.554 (0.066) | 0.900 (0.004) |
| Male | 4.175 (0.069) | 5.222 (0.070) | 0.728 (0.016) | 3.029 (0.069) | 3.764 (0.077) | 0.897 (0.004) |

The prediction metrics between predicted brain-age and chronological age for UK Biobank: $R^2$, root mean square error (RMSE), and mean absolute error (MAE) for the train data using 10-fold cross validation with 10 repetitions per fold, given as mean (standard deviation).
Subindex *orig* refers to values before bias correction.
Subindex *corr* refers to values after bias correction.

**Appendix 1—table 5.** Prediction metrics for females and males in testing set.

Pooled cohorts

|  | MAE | RMSE | $R^2$ |
|---|---|---|---|
| Female | 5.481 | 6.013 | 0.316 |
| Male | 6.201 | 6.217 | 0.364 |

The prediction metrics before bias correction between predicted brain-age and chronological age for the testing cohorts pooled together: $R^2$, root mean square error (RMSE), and mean absolute error (MAE) for females and males.

**Appendix 1—table 6.** Comparison of prediction's metrics between females and males.

|  | Fisher's z | P-Value |
|---|---|---|
| Females vs males | 1.542 | 0.123 |

Testing whether the Pearson's correlation coefficients from the brain-age prediction against the chronological age is similar between males and females pooled from all independent cohorts, via Fisher transformation.

**Appendix 1—table 7.** Prediction metrics for all independent cohorts using aging signature.

Aging Signature – performance for brain age prediction

|  | Correlation with age |  | MAE (y) | $R^2$ | RMSE |
|---|---|---|---|---|---|
| Cohorts |  |  |  |  |  |
| ALFA+ | −0.230 | <0.001 | 3.83 | 0.06 | 4.61 |
| ADNI | −0.256 | <0.001 | 4.60 | 0.07 | 5.85 |
| EPAD | −0.412 | <0.001 | 5.26 | 0.17 | 6.43 |
| OASIS | −0.455 | <0.001 | 6.64 | 0.21 | 8.41 |

We tested the linear association of the widely-used neuroanatomical aging signature (***Bakkour et al., 2013***) with chronological age, to compare its performance with the XGboost brain-age prediction. The aging signature is a map of specific brain regions that undergo cortical atrophy in normal aging. Our brain-age estimation outperformed the aging signature (Pearson's r [William's test], P<0.001; RMSE [F-test] P<0.001, for all cohorts). These analyses were computed on the Pearson's correlation coefficient (R) between predicted brain-age and chronological age, $R^2$, root mean square error (RMSE), and mean absolute error (MAE) for each of the independent cohorts after bias correction.

**Appendix 1—table 8.** Interaction affects between age and sex effects for each signiciant SHAP-selected ROI.

| Cortical thickness | | Subcortical Volumes | | Cortical Volumes | |
|---|---|---|---|---|---|
| ROI | P>|t| | ROI | P>|t| | ROI | P>|t| |
| L_frontalpole | <0.0001 | 3rd-Ventricle | <0.0001 | L_entorhinal | 0,019 |
| L_inferiorparietal | 0,104 | 4th-Ventricle | 0,976 | L_insula | <0.0001 |
| L_isthmuscingulate | <0.0001 | Brain-Stem | <0.0001 | L_isthmuscingulate | <0.0001 |
| L_lateraloccipital | 0,235 | CC_Anterior | <0.0001 | L_middletemporal | <0.0001 |
| L_lateralorbitofrontal | <0.0001 | CC_Central | 0,828 | L_parsopercularis | <0.0001 |
| L_middletemporal | 0,285 | CC_Mid_Posterior | 0,685 | L_parsorbitalis | <0.0001 |
| L_paracentral | 0,08 | CC_Posterior | 0,024 | L_parstriangularis | <0.0001 |
| L_parstriangularis | 0,014 | CSF | 0,136 | L_pericalcarine | 0,007 |
| L_precentral | <0.0001 | Left-Accumbens-area | <0.0001 | L_rostralmiddlefrontal | <0.0001 |
| L_precuneus | 0,007 | Left-Amygdala | <0.0001 | L_superiorfrontal | <0.0001 |
| L_superiorfrontal | <0.0001 | Left-Caudate | <0.0001 | R_entorhinal | 0,006 |

*Appendix 1—table 8 Continued on next page*

*Appendix 1—table 8 Continued*

| Cortical thickness | | Subcortical Volumes | | Cortical Volumes | |
|---|---|---|---|---|---|
| L_superiortemporal | 0,071 | Left-Cerebellum-Cortex | <0.0001 | R_fusiform | <0.0001 |
| L_transversetemporal | <0.0001 | Left-Cerebellum-White-Matter | <0.0001 | R_insula | <0.0001 |
| R_caudalanteriorcingulate | 0,452 | Left-Hippocampus | <0.0001 | R_middletemporal | <0.0001 |
| R_cuneus | 0,235 | Left-Inf-Lat-Vent | <0.0001 | R_parsorbitalis | <0.0001 |
| R_frontalpole | <0.0001 | Left-Lateral-Ventricle | <0.0001 | R_parstriangularis | <0.0001 |
| R_inferiorparietal | 0,379 | Left-Putamen | <0.0001 | R_pericalcarine | 0,258 |
| R_lateraloccipital | 0,212 | Left-Thalamus-Proper | <0.0001 | R_postcentral | <0.0001 |
| R_lateralorbitofrontal | <0.0001 | Left-VentralDC | <0.0001 | R_rostralmiddlefrontal | <0.0001 |
| R_lingual | 0,789 | Left-choroid-plexus | <0.0001 | R_superiorfrontal | <0.0001 |
| R_middletemporal | 0,205 | Optic-Chiasm | <0.0001 | R_supramarginal | <0.0001 |
| R_paracentral | 0,301 | Right-Amygdala | <0.0001 | R_transversetemporal | <0.0001 |
| R_parstriangularis | 0,088 | Right-Caudate | <0.0001 | | |
| R_pericalcarine | 0,086 | Right-Cerebellum-Cortex | <0.0001 | | |
| R_posteriorcingulate | 0,003 | Right-Cerebellum-White-Matter | <0.0001 | | |
| R_precentral | <0.0001 | Right-Hippocampus | <0.0001 | | |
| R_precuneus | 0,215 | Right-Inf-Lat-Vent | <0.0001 | | |
| R_rostralmiddlefrontal | <0.0001 | Right-Lateral-Ventricle | <0.0001 | | |
| R_superiorfrontal | 0,003 | Right-Pallidum | <0.0001 | | |
| R_superiortemporal | 0,183 | Right-Putamen | <0.0001 | | |
| R_transversetemporal | <0.0001 | Right-Thalamus-Proper | <0.0001 | | |
| | | Right-VentralDC | <0.0001 | | |
| | | Right-choroid-plexus | <0.0001 | | |

**Appendix 1—table 9.** Comparison of brain-age deltas for the different amyloid-β, AT and *APOE* status.

| | Mean square | F | *P*-Value |
|---|---|---|---|
| **CU** | | | |
| Amyloid-β status | 35.787 | 39.369 | **<0.001** |
| AT Stage | 21.550 | 11.626 | **<0.001** |
| *APOE* status | 16.218 | 5.862 | **<0.001** |
| **MCI** | | | |
| Amyloid-β status | 47.166 | 52.412 | **<0.001** |
| AT Stage | 50.064 | 27.701 | **<0.001** |
| *APOE* status | 10.049 | 3.425 | **0.017** |

Brain-age deltas for pooled CU and MCI individuals were compared for the different amyloid-β, AT and *APOE* status with ANCOVA models adjusted by age and sex.

Amyloid-β status was defined by CSF (ALFA+, ADNI and EPAD) or amyloid PET (OASIS).

Significant values (*P*<0.05) are marked in bold.

APOE, apolipoprotein E; ANCOVA, analysis of covariance.

**Appendix 1—table 10.** Mean brain-age delta values for the different amyloid-β, AT and *APOE* status.

| | | CU brain-age delta, M (95% CI) | MCI brain-age delta, M (95% CI) |
|---|---|---|---|
| Amyloid-β status | Aβ- | –0.468 (-0.708,–0.228) | –1.618 (-2.231,–1.005) |
| | Aβ+ | 0.802 (0.485, 1.119) | 1.033 (0.577, 1.490) |
| | A-T- | –0.214 (-0.507, 0.078) | –1.623 (-2.239,–1.007) |
| | A+T- | 0.916 (0.506, 1.325) | 0.561 (-0.066, 1.188) |
| AT Stage | A+T + | 0.945 (0.140, 1.750) | 1.410 (-0.753, 2.066) |
| | *APOE*-ε2 | –0.726 (-1.298,–0.153) | –0.660 (-2.234, 0.914) |
| | *APOE*-ε33 | –0.225 (-0.506, 0.056) | –0.522 (-1.0750, 0.031) |
| | *APOE*-ε4 | 0.503 (0.193, 0.813) | 0.617 (0.050, 1.184) |
| *APOE* status | *APOE*-ε24 | –0.092 (-1.109, 0.924) | 1.257 (-1.067, 3.581) |

Notes: brain-age delta are expressed as mean (M) and 95% confidence interval (CI).

Amyloid-β status was defined by CSF (ALFA+, ADNI and EPAD) or amyloid PET (OASIS). Abbreviations: APOE, apolipoprotein E.

**Appendix 1—table 11.** Relationships between validation variables and brain-age delta across independent cohorts.

| Model | | β | SE | P-Value | [0.025] | [0.975] | N | Effect size |
|---|---|---|---|---|---|---|---|---|
| | | | | ALFA+ | | | | |
| Amyloid-β pathology | | 0.796 | 0.377 | **0.035** | 0.056 | 1.537 | 355 | 0.217 |
| Amyloid-β / Tau pathology (ref: A-T-) | A+T- | 1.028 | 0.409 | **0.013** | 0.222 | 1.832 | 355 | 0.225 |
| | A+T + | 0.855 | 0.929 | 0.358 | –0.973 | 2.682 | 355 | 0.225 |
| | *APOE*-ε2 | –1.278 | 0.780 | 0.102 | –2.811 | 0.255 | | 0.325 |
| APOE status (ref: *APOE*-e33) | *APOE*-ε4 | 0.002 | 0.397 | 0.996 | –0.231 | 0.817 | | 0.001 |
| | *APOE*-ε24 | –0.294 | 1.273 | 0.817 | –2.797 | 2.797 | 355 | 0.075 |
| WMH | | 0.724 | 0.236 | **0.002** | 0.260 | 1.187 | 337 | 0.028 |
| CSF NfL | | 0.106 | 0.199 | 0.593 | –0.284 | 0.497 | 357 | 0.001 |
| Plasma NfL | | 0.116 | 0.217 | 0.594 | –0.311 | 0.543 | 343 | 0.001 |
| Aging signature change | | 0.082 | 0.234 | 0.728 | –0.378 | 0.542 | 236 | 0.001 |
| | | | | ADNI | | | | |
| Amyloid-β | | 0.182 | 0.525 | 0.730 | –0.852 | 1.216 | 233 | 0.047 |
| Amyloid-β / Tau pathology (ref: A-T-) | A+T- | –0.375 | 0.620 | 0.546 | –1.597 | 0.847 | 232 | 0.096 |
| | A+T + | 0.855 | 0.711 | 0.230 | –0.546 | 2.255 | 232 | 0.225 |
| | *APOE*-ε2 | 0.525 | 0.801 | 0.513 | –1.055 | 2.104 | | 0.142 |
| APOE status (ref: *APOE*-e33) | *APOE*-ε4 | 0.137 | 0.579 | 0.813 | –1.003 | 1.278 | | 0.034 |
| | *APOE*-ε24 | –1.183 | 2.811 | 0.674 | –6.722 | 4.356 | 233 | 0.000 |
| WMH | | 0.483 | 0.243 | **0.040** | 0.004 | 0.963 | 218 | 0.018 |
| CSF NfL | | 0.707 | 0.784 | 0.380 | –0.947 | 2.361 | | 0.048 |
| Plasma NfL | | 0.195 | 0.331 | 0.557 | –0.459 | 0.849 | 165 | 0.002 |
| Aging signature change | | 0.343 | 0.713 | 0.634 | –1.108 | 1.794 | | 0.007 |
| | | | | EPAD | | | | |
| Amyloid-β | | 1.410 | 0.311 | **<0.001** | 0.800 | 2.020 | 601 | 0.377 |

*Appendix 1—table 11 Continued on next page*

*Appendix 1—table 11 Continued*

| Model | | β | SE | *P*-Value | [0.025] | [0.975] | N | Effect size |
|---|---|---|---|---|---|---|---|---|
| Amyloid-β / Tau pathology (ref: A-T-) | A+T- | 1.109 | 0.348 | **0.002** | 0.474 | 2.719 | 575 | 0.298 |
| | A+T + | 1.59 | 0.572 | **0.005** | 0.474 | 2.719 | 575 | 0.417 |
| APOE status (ref: *APOE*-e33) | *APOE*-ε2 | −0.318 | 0.573 | 0.573 | −1.555 | 0.579 | | 0.008 |
| | *APOE*-ε4 | 0.292 | 0.340 | 0.390 | −0.375 | 0.959 | | 0.076 |
| | *APOE*-ε24 | −0.506 | 0.807 | 0.531 | −2.090 | 1.078 | 601 | 0.131 |
| WMH | | 0.783 | 0.182 | **<0.001** | 0.426 | 1.140 | 417 | 0.045 |
| **OASIS** | | | | | | | | |
| Amyloid-β | | 0.917 | 0.441 | **0.038** | 0.051 | 1.784 | 407 | 0.246 |
| APOE status (ref: *APOE*-e33) | *APOE*-ε2 | −0.228 | 0.532 | 0.668 | −1.272 | 0.817 | | 0.060 |
| | *APOE*-ε4 | 1.238 | 0.419 | **0.003** | 0.415 | 2.061 | | 0.335 |
| | *APOE*-ε24 | 0.412 | 1.086 | 0.702 | −1.719 | 2.550 | 407 | 0.111 |

Notes: Relationships between validation variables and brain-age delta for all CU subjects from each cohort. Results given by the linear model: brain-age delta ~each variable +chronological age+sex. The regression coefficients (β), standard errors (SE), *P*-value, 95% Confidence Interval, number of individuals (N) and effect size are depicted for each variable.

Significant values (*P*<0.05) are marked in bold.

Effect size in categorical variables was calculated as Cohen's D, while Cohens f2 was calculated for continuous measurements.

Amyloid-β status was defined by CSF (ALFA+, ADNI and EPAD) or amyloid PET (OASIS).

APOE, apolipoprotein E; WMH, White Matter Hyperintensities; CSF, cerebrospinal fluid; NfL, neurofilament light.

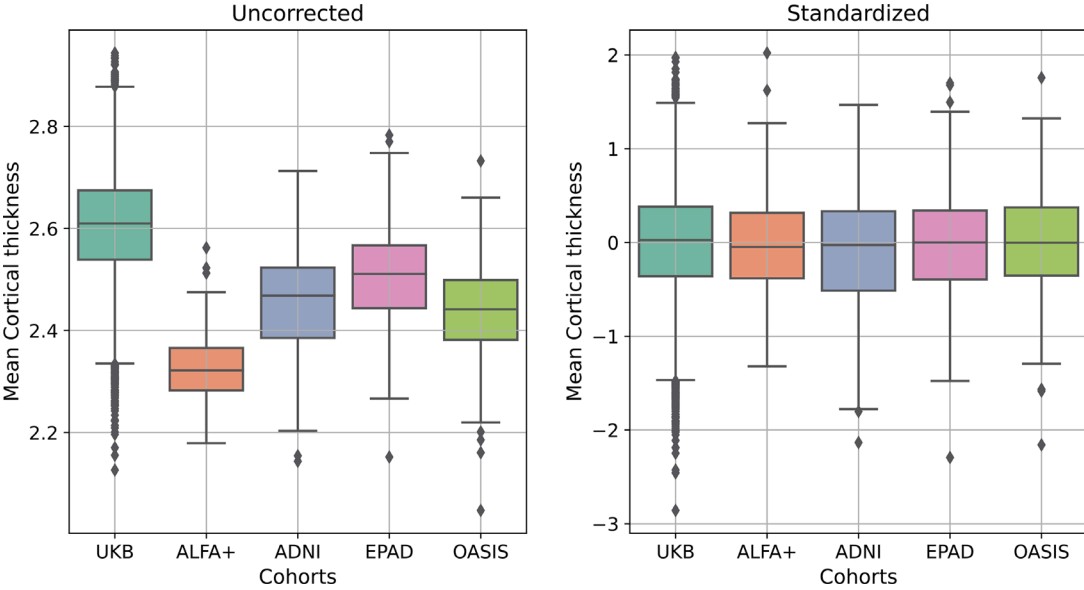

**Appendix 1—figure 1.** Median (and interquartile range) cortical thickness for all individuals from all cohorts (UK Biobank (UKB), ALFA+, ADNI, EPAD and OASIS), without any correction (left) and after the standardization procedure (right). The sample size can be seen in *Table 1*.

## Secondary analysis: Selection of Extreme Subjects
### Methods
The cohorts included in this study include mainly CU individuals with mean age 65–70 years and ranges of around 10 years. Therefore, we expected to find small differences in the biomarkers' values used as validation measurements in these subjects. In addition, the effects of these biomarkers on the brain-age should be subtle, as many other processes and factors can influence the predicted brain age. Therefore, we wanted to identify the individuals whose predicted brain-age deviate the most from chronological aging, i.e., individuals with the highest positive or lowest negative brain-age deltas, to study the mentioned associations. With this aim, we selected the individuals whose brain-age delta was included within the 10th and 90th percentile of the distribution for each independent

cohort. We then pooled together these selected individuals, who will be referred from now on as extreme subjects with decelerated brain-aging and accelerated brain-aging.

Once selecting these extreme subjects, we performed between groups comparison for the clinical and validation measurements, between the decelerated brain aging subjects and the accelerated brain aging subjects. For this we performed independent t tests or Mann–Whitney U tests for continuous variables and chi-square tests for categorical variables. We also tested for interactions between age and the validation variables on brain-age delta. Lastly, we stratified the decelerated and the accelerated brain aging subjects by sex and performed the same between groups comparison to study the sex differences of the composition of the demographics and validation variables found in the decelerated and accelerated brain-aging subjects. We also tested for interactions between sex and the validation variables on brain-age delta.

## Results

Finally, we investigated whether individuals with extreme brain age phenotypes differed in their demographical and clinical characteristics as well as in the AD and neurodegeneration-related biomarkers and risk factors. The extreme phenotypes were defined as those individuals at the 10th percentile of brain-age delta (henceforth, decelerated brain aging) and 90th percentile of brain-age delta (accelerated brain aging). The extreme groups were defined, firstly, including both sexes and, secondly, separately in females and males.

Across the four cohorts, including both sexes, we found that Mini-Mental State Examination (MMSE) was significantly lower in the accelerated brain aging group for both the CU and the MCI individuals (CU: $P=0.006$, D=0.300 and MCI: $P<0.001$, D=0.883). Years of education, percentage of females and males, and chronological age were not significantly different between the extreme groups.

For both CU and MCI, in the accelerated brain aging group, we found a higher proportion of Aβ-positive individuals (CU: $P<0.001$, V=0.231 and MCI: $P<0.001$, V=0.502), more advanced AT stages (CU: $P<0.001$, V=0.239 and MCI: $P<0.001$, V=0.531), and larger WMH total volume (CU: $P<0.001$, D=0.707 and MCI: $P=0.044$, D=0.430). In the MCI group, a higher proportion of *APOE-ε4* carriers in the accelerated brain aging group was found with respect to the decelerated brain aging group ($P<0.001$, D=0.455). When studying the differences of neurodegeneration markers across the two extreme groups, we found significantly higher plasma NfL in the accelerated brain aging group only among MCI individuals (MCI: $P=0.007$, D=0.696). In addition, we found a significant interaction effect of age and plasma NfL on CU brain-age delta ($P_{interaction} = 0.034$) within the CU individuals, by which the measures of plasma NfL were higher with age in the accelerated brain-aging group. CSF NfL was only available in the ALFA +cohort (N=82) and a similar trend to that of plasma NfL was observed, but without reaching statistical significance.

We next compared the extreme brain age phenotypes calculated separately for CU females and males. In both CU females and males defined extreme groups, we found higher proportion of Aβ-positive individuals, higher proportion of more advanced AT stages, larger WMH and lower aging signature values (that is, reduced cortical thickness in aging-vulnerable regions) in the accelerated brain aging groups. Effect sizes showed that the increased WMH in the accelerated brain aging group was larger in females than in males. On the contrary, the effect of the proportion of Aβ-positive individuals in the accelerated brain aging group was stronger in males than in females. We also found lower MMSE scores in the CU accelerated brain aging males. In addition, we found a significant interaction effect of sex and plasma NfL on CU brain-age delta ($P_{interaction} = 0.038$) within the CU extremes individuals, by which the measures of plasma NfL were larger in females in the accelerated brain-aging group.

