## [Editor Report]

The study has significance for the field of dementia research and neurodegenerative diseases more broadly. Using the brain-age paradigm, the main findings are that having an older-appearing brain is associated with more advanced stages of amyloid and tau pathology, higher white matter hyperintensities, higher plasma NfL and carrying the APOE-e34 allele. Findings were broadly similar in cognitively normal people and people with mild cognitive impairment and there is also some evidence for sex differences.

---

## [Decision Letter]

**Decision letter after peer review:**

Thank you for submitting your article "Prediction Using Machine Learning on Structural Neuroimaging Data: Multi-Cohort Validation Against Biomarkers of Alzheimer's Disease and Neurodegeneration stratified by sex" for consideration by *eLife*. Your article has been reviewed by 3 peer reviewers, and the evaluation has been overseen by a Reviewing Editor and Jeannie Chin as the Senior Editor. The following individuals involved in the review of your submission have agreed to reveal their identity: Rory Boyle (Reviewer #2); James Cole (Reviewer #3).

Essential revisions:

1) Given the large number of tests conducted, the lack of multiple comparison correction means that some apparently significant associations may be statistically inflated.

2) Claims regarding sex differences should be tempered to be more in line with the strength of evidence.

3) Reviewers noted that brain-age δ was not associated with longitudinal brain change and that the current study is solely a cross-sectional analysis. The implications for this should be discussed – specifically, whether it indicates that brain-age δ does not reflect accelerated brain aging but instead early life factors.

*Reviewer #1 (Recommendations for the authors):*

– ADNI dataset includes participants with a dementia diagnosis and many of those are in the AD continuum. What was the reason for not including those cases in this study?

– When correcting brain-age estimates for bias, the bias regression line was estimated for each cohort separately. I believe cohort refers to CU vs MCI cases in this context. Is there a rationale for expecting disease stage-dependent bias in brain-age estimates?

– Please clarify the use of the term 'cohort'. does it refer to CU vs MCI diagnostic groups or does it refer to different datasets? if lateral, then I think certain analyses should have been performed by cohort and diagnostic groups since the brain age behavior might be different at different disease stages. for instance, since UKBioBank is all CU participants, it is highly likely that the model performance within ADNI CU and within ADNI MCI cases might be different in terms of R2 and MAE.

– Once the brain-age estimates are corrected for bias based on regression against chronological age, including age as a covariate in linear regression models for each validation variable might lead to double correction for age bias.

– In reference to the paragraph: "We also studied the differences in volumes and cortical thickness between females and males in the UKBioBank for the brain regions that contributed the most to the prediction according to the SHAP values. With this aim we performed regression models for each ROI with sex as a predictor variable, in which linear and quadratic expansions of age, site, and TIV (only included for volume ROIs), were included as covariates.", it is not clear why this regression modeling was necessary if SHAP was used to identify regions contributing the most to brain age prediction. Furthermore, ROI volumes were initially residualized for site and TIV and similarly, thickness values were residualized for site effects. If that's the case, why include site and TIV as a covariate in this analysis? Also, this is the only analysis that higher order age associations were considered. What was the rationale for pursuing non-linear age associations in this case but not in other models?

*Reviewer #2 (Recommendations for the authors):*

This is very nice work but I think that the findings could be really strengthened by addressing the limitations highlighted in the public review. I can definitely appreciate the impressive amount of work that has already gone into this study and I do not lightly ask for additional work. My concerns would be addressed by correcting your findings for multiple comparisons, re-interpreting your results after correction for multiple comparisons, and discussing the other limitations in-text that can address my concerns.

I have two other main concerns that can be addressed by text edits. First, the novelty of this work is overstated and does not fairly represent the brain-age δ literature. The authors incorrectly state that "there are no comprehensive studies validating this measurement in association with specific biological markers of AD pathology (i.e. Amyloid-B [Ab] and tau pathology), neurodegeneration and cerebrovascular disease. Various studies have reported associations between brain-age δ and biomarkers of amyloid-B, tau, neurodegeneration, and cerebrovascular disease (Cole et al., 2017 Neurobiology of Aging; Huang et al., 2021 Radiology Artificial Intelligence; Millar et al., 2022 bioRxiv; Popescu et al., 2020 Human Brain Mapping; Wagen et al., 2022 Lancet Healthy Longevity). Therefore, this particular emphasis on novelty is not correct and is not representative of the literature. Can the authors please de-emphasize this novelty and acknowledge the good work carried out by other researchers that have addressed some of these questions previously?

Second, "trend" or "trending" is used in the manuscript to refer to findings that are nearly statistically significant. For example, see Lines 333-336, Lines 291-294, and Line 444. However, this is not a meaningful description of a statistical result as we do not know which way these results are 'trending'. It can give the impression that the results are likely significant but did not reach significance for some unstated reason. However, the distance between the sex*plasma interaction in CU (P=.092) than reaching statistical significance (.092 –.05 = .042) is further than the distance of the significant result for APOE-e4 carriers vs APOE-e33 carriers of P = .032 reaching 'non-significance' (.05 –.032 = .018). Likewise for the A+T+*sex interaction (P=.071). In that sense, if you are using language such as the trend to describe findings, you could equally use it to describe findings showing a trend towards non-significance. As such, please avoid using this language and state your results as they are.

References:

Cole et al., 2017 Neurobiology of Aging: https://doi.org/10.1016/j.neurobiolaging.2017.04.006

Huang et al., 2021 Radiology Artificial Intelligence: https://doi.org/10.1148/ryai.2021200171

Millar et al., 2022 bioRxiv: https://doi.org/10.1101/2022.08.25.505251

Popescu et al., 2020 Human Brain Mapping: https://doi.org/10.1002/hbm.25133

Wagen et al., 2022 Lancet Healthy Longevity: https://doi.org/10.1016/S2666-7568(22)00167-2

Sanford et al., 2022, Human Brain Mapping: https://doi.org/10.1002/hbm.25983

Subpramaniapillai et al., 2021 NeuroImage Clinical: https://doi.org/10.1016/j.nicl.2021.102620

Vidal-Pineiro et al. 2021 *eLife*: https://doi.org/10.7554/*eLife*.69995

*Reviewer #3 (Recommendations for the authors):*

– Consider the impact that the number of tests run has on the interpretation.

– It was not entirely clear from the Methods how MCI was defined in each cohort. If these definitions are not consistent between cohorts, this should be highlighted as a limitation.

– It is inappropriate to report accuracy performance metrics after correction for brain-age bias (e.g., Table 3). The point of the correction process is not to improve model performance post hoc, but instead to improve the interpretability of the metrics by removing residual correlation with age. All performance metrics 'after bias correction' should be removed from the manuscript

– Figure 4. It was not obvious what the interactions related to. Of the 9 interactions reported, one is by sex, the other by age. Also, why were the points grouped and coloured by positive versus negative brain-age δ? Generally, this figure could be improved for the sake of clarity.

– Why was the 'aging signature' included? Please provide stronger motivation for why this was relevant.

– The analysis on the top and bottom 10% of participants were not well motivated. Why include this when regression models have already by used that assess 100% of the data, instead of 20%? It was not clear where these results were reported either, so I would recommend omitting this analysis.

– Avoid drawing interpretations that males and females are different in cases where there is no statistical test of this hypothesis. Being non-identical does not necessarily mean that sex differences are statistically significant (e.g., Figure 1d).

– I recommend reporting confidence intervals whenever you report an effect size. This facilitates the comparison of effect sizes between tests.

---

## [Author Response]

Essential revisions:1) Given the large number of tests conducted, the lack of multiple comparison correction means that some apparently significant associations may be statistically inflated.

The reviewer’s suggestion is well received. To address this issue, we have now corrected p-values by the False Discovery Rate (FDR) to control the overall type I error rate in the multiple comparisons made in the study.

2) Claims regarding sex differences should be tempered to be more in line with the strength of evidence.

We agree with the reviewer’s comment that sex differences should be better characterized in the manuscript. We have revised our language to accurately reflect the strength of evidence for these findings. Examples of this are shown below:

Discussion, page 21, lines 477-479: “Our results also indicate that there might be sex differences in the development of brain aging trajectories, which must be further characterized.”

Discussion, page 23, lines 609-611: “However, the lack of significant brain-age δ-by-sex interactions in our analyses reflect the limitations in the evidence for the above-mentioned sex-differences. Further work exploring these sex differences in detail is warranted”.

3) Reviewers noted that brain-age δ was not associated with longitudinal brain change and that the current study is solely a cross-sectional analysis. The implications for this should be discussed – specifically, whether it indicates that brain-age δ does not reflect accelerated brain aging but instead early life factors.

The reviewer raises an important point. We did not expect the brain-age δ to have a strong impact on longitudinal brain changes, given the short follow-up time (3 years, approximately) which is comparable to the observed variability of our brain-age δ in the cross-sectional analysis. By definition, the cross-sectional analysis reflects the accumulated effect of biological factors on brain aging for about 60 years (the average age of our samples).

We agree with the reviewer that these points deserved further attention in the discussion. In light of this, we have added a discussion in the manuscript’s main text to address the implications of the longitudinal results. We have considered potential explanations for the findings, including the possibility that brain-age δ reflects early life factors rather than accelerated brain aging (as previous literature has suggested) and the possibility that a 3-year interval may not be sufficient for brain-age δ to capture longitudinal changes. We also note that brain-age δ is expected to account for small variability in the processes that contribute to aging and that a longer time difference may be needed to detect longitudinal changes. Examples of this are shown below:

Discussion, page 21, lines 475-477: “However, our results do not show that variation in cross-sectional brain-age δ was able to capture longitudinal atrophy and, therefore, it is solely a cross-sectional analysis.”

Discussion, page 21, lines 482-483: “The capacity of brain-age δ to de detect accelerated aging must be further studied.”

Discussion, page 22, lines 530-537: “Conversely, brain-age δ was not associated with longitudinal aging signature change. This might indicate, as previously shown (Vidal-Pineiro et al., 2020), that brain-age δ might not reflect accelerated brain aging but instead early life factors. However, the lack of this association might also be influenced by the short amount of time between time visits (average of 3 years) or the measurement used to measure longitudinal atrophy (aging signature). Therefore, further research is needed to determine the relative contribution of early life factors and accelerated aging to brain-age δ estimates.”

Reviewer #1 (Recommendations for the authors):– ADNI dataset includes participants with a dementia diagnosis and many of those are in the AD continuum. What was the reason for not including those cases in this study?

We thank the reviewer for this relevant question. Estimates of brain-age have been described previously in AD dementia patients (Kaufman, 2019; Beheshti, 2918). The main novelty of our study is to assess the impact of abnormal AD biomarkers and risk factors on brain-age estimates. We believe that these are particularly informative in preclinical and prodromal AD stages in order to potentially inform therapeutic interventions.

We have now made this point explicit in:

Introduction, Page 4, lines 154-158: “Nonetheless, there remains a need to replicate some of these results and to study the associations between brain-age prediction and abnormal biomarkers of AD and neurodegeneration in preclinical and prodromal AD stages in different and independent cohorts and in a larger sample size. These results would be particularly informative to potentially inform therapeutic interventions.”

Materials and methods, Page 25, lines 676-679: “We did not include individuals with a dementia diagnosis in the AD continuum because we wanted to focus on assessing the impact of abnormal AD biomarkers and risk factors on brain-age estimates in preclinical and prodromal AD stages.”

– When correcting brain-age estimates for bias, the bias regression line was estimated for each cohort separately. I believe cohort refers to CU vs MCI cases in this context. Is there a rationale for expecting disease stage-dependent bias in brain-age estimates?

We thank the reviewer for this question. We would first like to clarify that the term 'cohort' in our study refers to the different datasets used in our analysis, rather than the diagnostic groups (CU vs MCI). Our study includes five distinct MRI datasets (UK BioBank, ALFA+, ADNI, EPAD and OASIS), and the term 'cohort' is used to distinguish between the five. We apologize for any confusion caused and appreciate your feedback. We have revised the text to make this distinction clearer, and hope that the revised explanation will clarify any confusion.

Materials and methods, Page 25, lines 670-675: “These cohorts included individuals with different diagnosis: CU and MCI. ADNI cohort included CU and MCI individuals from ADNI 1,2 and 3 (*N =* 751, CU = 253, MCI = 498), the EPAD cohort included CU and MCI individuals (*N =* 808, CU = 653, MCI = 155), the ALFA+ cohort included only CU individuals (*N =* 380) and the OASIS cohort included CU individuals (*N* = 407).”

Regarding the potential disease-stage bias in brain-age estimates, we agree with the reviewer that we did not expect a significant effect. We have actually assessed the regressions between brain-age and chronological age and have not found any differences in the slopes. Of course, we find different intercepts due to the fact that MCI patients show an ‘older-appearing’ brain due to the pathologic process with respect to healthy controls.

We hope this clarifies our rationale and appreciate the opportunity to address this issue.

– Please clarify the use of the term 'cohort'. does it refer to CU vs MCI diagnostic groups or does it refer to different datasets? if lateral, then I think certain analyses should have been performed by cohort and diagnostic groups since the brain age behavior might be different at different disease stages. for instance, since UKBioBank is all CU participants, it is highly likely that the model performance within ADNI CU and within ADNI MCI cases might be different in terms of R2 and MAE.

Thank you for your comment regarding the use of the term 'cohort' in our manuscript and your suggestions. The term 'cohort' in our study refers to the different datasets used in our analysis, rather than the diagnostic groups (CU vs MCI). Our study includes five distinct MRI datasets (UK BioBank, ALFA+, ADNI, EPAD and OASIS), and the term 'cohort' is used to distinguish between the five. Within two specific cohorts (ADNI and EPAD) we have two diagnostic groups (CU and MCI). We have revised the text to make this distinction clearer, and hope that the revised explanation will clarify any confusion.

To address the reviewer’s point, we have studied whether the brain age behavior is different at different disease stages in these cohorts. We have now included a Table in Supplementary Material (Supplemental Table S3) to show the model performance separately for CU and MCI individuals in the cohorts containing both diagnostic groups. The results show that there are no substantial differences in the behavior of brain age across cohorts and diagnostic groups. We believe that this result reinforces the robustness of the brain-age estimates across cohorts and diagnostic conditions. We hope that this will address the reviewer’s concerns.

– Once the brain-age estimates are corrected for bias based on regression against chronological age, including age as a covariate in linear regression models for each validation variable might lead to double correction for age bias.

We thank the reviewer for this insightful comment regarding the correction of brain-age estimates.

It has previously been shown in the literature that: if the estimated brain-age δ is not orthogonal to age, and if the other variables for which the association with brain-age δ are being studied have not been deconfounded with respect to age, then any associations between brain-age δ and these variables might be more driven by age and not brain-age δ (Smith 2019; de Lange and Cole, 2020). Therefore, to overcome this, the two-step procedure for correcting age bias that we have implemented has been previously proposed and used in previous studies in the field of brain-age δ (Smith 2019; de Lange and Cole, 2020; de Lange 2020). Therefore, we believe that, in order to account for all age-related bias, it is important to include age as a covariate in the subsequent regression models for each validation variable. We hope that this revised explanation will address the concerns you have raised.

– In reference to the paragraph: "We also studied the differences in volumes and cortical thickness between females and males in the UKBioBank for the brain regions that contributed the most to the prediction according to the SHAP values. With this aim we performed regression models for each ROI with sex as a predictor variable, in which linear and quadratic expansions of age, site, and TIV (only included for volume ROIs), were included as covariates.", it is not clear why this regression modeling was necessary if SHAP was used to identify regions contributing the most to brain age prediction. Furthermore, ROI volumes were initially residualized for site and TIV and similarly, thickness values were residualized for site effects. If that's the case, why include site and TIV as a covariate in this analysis? Also, this is the only analysis that higher order age associations were considered. What was the rationale for pursuing non-linear age associations in this case but not in other models?

We thank the reviewer for raising this point. Please note that the aim of this analysis was not to identify regions contributing the most to brain age but to describe sex differences in brain regions contributing the most to brain-age models for males and females.

With respect to the inclusion of higher-order associations with age in this analysis, it was initially motivated by previous literature showing non-linear effects. However, we agree with the reviewer that it is inconsistent with the rest of the analysis in the manuscript to include non-linear associations in this analysis. Therefore, we have now only included linear ones.

We have modified the main text accordingly to introduce the revised changes and to clarify the aim of this secondary analysis further:

Materials and methods, Page 30, lines 956-960: “We also performed two different secondary and exploratory analyses. First, we assessed whether the brain regions (volumes and cortical thickness) that contributed the most to the prediction of females and males in the UKBioBank were different for each sex. With this aim we performed regression models to study the interaction effect of sex for each of the selected ROIs by including a sex-age interaction term.”

Reviewer #2 (Recommendations for the authors):This is very nice work but I think that the findings could be really strengthened by addressing the limitations highlighted in the public review. I can definitely appreciate the impressive amount of work that has already gone into this study and I do not lightly ask for additional work. My concerns would be addressed by correcting your findings for multiple comparisons, re-interpreting your results after correction for multiple comparisons, and discussing the other limitations in-text that can address my concerns.

Thank you for your positive feedback on our work and for bringing to our attention the limitations that need to be addressed in order to strengthen our findings. We appreciate your constructive criticism and value your insights.

We have taken your comments into consideration and have performed corrections for multiple comparisons using FDR, re-interpreted our results after correction for multiple comparisons, and discussed the other limitations in the text. We believe that these changes will address your concerns and strengthen the validity of our findings.

I have two other main concerns that can be addressed by text edits. First, the novelty of this work is overstated and does not fairly represent the brain-age δ literature. The authors incorrectly state that "there are no comprehensive studies validating this measurement in association with specific biological markers of AD pathology (i.e. Amyloid-B [Ab] and tau pathology), neurodegeneration and cerebrovascular disease. Various studies have reported associations between brain-age δ and biomarkers of amyloid-B, tau, neurodegeneration, and cerebrovascular disease (Cole et al., 2017 Neurobiology of Aging; Huang et al., 2021 Radiology Artificial Intelligence; Millar et al., 2022 bioRxiv; Popescu et al., 2020 Human Brain Mapping; Wagen et al., 2022 Lancet Healthy Longevity). Therefore, this particular emphasis on novelty is not correct and is not representative of the literature. Can the authors please de-emphasize this novelty and acknowledge the good work carried out by other researchers that have addressed some of these questions previously?

We have re-evaluated our claims regarding the novelty of our study and have revised our language to de-emphasize this aspect and acknowledge the good work carried out by other researchers that have addressed some of these questions previously. We also would like to emphasize that our research question focuses on studying the association of fluid and imaging biomarkers of AD pathology neurodegeneration and cerebrovascular disease in individuals that are in the preclinical stages of AD. Taking this into consideration, we have made the necessary additions in the main text:

The two above-mentioned works that are in line with our research question (Millar et al., 2022 bioRxiv; Wagen et al., 2022 Lancet Healthy Longevity), were not publicly available when we submitted our manuscript. Thanks to the reviewer, we have now included these references in the text (Introduction, Page 3, lines 135-142):

“Recent literature has studied the associations between brain-age δ and the above-mentioned biomarkers with different aims. Brain-age δ was recently associated with plasma biomarkers of neurodegeneration and with imaging biomarkers of cerebrovascular disease in the 1946 british Birth Cohort (Wagen et al., 2022), however it was not associated with CSF biomarkers of neurodegeneration (Millar et al., 2022). Moreover, although brain-age δ was significantly associated with AD biomarkers of amyloid and tau in MCI individuals, these associations were not found in CU individuals (Millar et al., 2022; Wagen et al., 2022).”

Regarding the two suggested references for Huang et al., 2021 Radiology Artificial Intelligence and Popescu et al., 2020 Human Brain Mapping, we know that these two works are very relevant to the field and well done. That is the reason why we had already mentioned Huang et al., 2021 Radiology Artificial Intelligence in the text: “A recent study used brain-age measurements to identify amnestic MCI (aMCI), the typical clinical presentation of prodromal AD, from other individuals with MCI, by studying the association with AD risk factors such as apolipoprotein E (APOE) and Aβ (Huang et al., 2021).”. Although these two works are very relevant, we wanted to emphasize that their research question was different from ours. They used brain-age δ in addition to other biomarkers to predict different disease stages or disease progression. Therefore, our mentioned original novelty was expressed considering that our work was different from theirs. However, now, thanks to the reviewer, we have also included Popescu et al., 2020 Human Brain Mapping in the text and we have de-emphasized the novelty: Introduction, page 3, lines 120-124:

“Even though these studies support the association of brain-age δ as a biomarker of biological aging with relevance to various brain diseases, it is of interest to further validate this measurement in association with specific biological markers of AD pathology (i.e. Amyloid-β [Aβ] and tau pathology), neurodegeneration and cerebrovascular disease in the earliest stages of AD. “

We didn’t originally include the work from Cole et al., 2017 Neurobiology of Aging, as it focuses on Down Syndrome individuals. However, now we have also mentioned their important and interesting work in the text: Introduction, page 4, lines 152-164:

“The association of brain-age with these biomarkers have also been shown in other diseases, by which brain-age was associated with A deposition in Down syndrome (Cole et al., 2017).”

Second, "trend" or "trending" is used in the manuscript to refer to findings that are nearly statistically significant. For example, see Lines 333-336, Lines 291-294, and Line 444. However, this is not a meaningful description of a statistical result as we do not know which way these results are 'trending'. It can give the impression that the results are likely significant but did not reach significance for some unstated reason. However, the distance between the sex*plasma interaction in CU (P=.092) than reaching statistical significance (.092 –.05 = .042) is further than the distance of the significant result for APOE-e4 carriers vs APOE-e33 carriers of P = .032 reaching 'non-significance' (.05 –.032 = .018). Likewise for the A+T+*sex interaction (P=.071). In that sense, if you are using language such as the trend to describe findings, you could equally use it to describe findings showing a trend towards non-significance. As such, please avoid using this language and state your results as they are.

We agree with the reviewer’s comment regarding the use of the term "trend" or "trending" to describe findings that are nearly statistically significant is imprecise and can lead to confusion. Therefore, we have now revised our language to state our results as they are, without relying on imprecise terminology. We believe that these revisions will ensure that our results are clearly and accurately represented in the manuscript.

Reviewer #3 (Recommendations for the authors):– Consider the impact that the number of tests run has on the interpretation.

We appreciate your attention to the impact of multiple tests on the interpretation of the results. To address this concern, we have carefully considered the number of tests run and have used appropriate methods to correct for multiple comparisons: we have used the Benjamini-Hochberg procedure to control the false discovery rate and ensure that the results are robust. Additionally, we have reported the corrected p-values and effect sizes, along with their associated confidence intervals, to provide a more comprehensive picture of the results.

We hope this information provides a better understanding of our methods and findings, and we look forward to your continued feedback.

– It was not entirely clear from the Methods how MCI was defined in each cohort. If these definitions are not consistent between cohorts, this should be highlighted as a limitation.

We thank the reviewer for this important comment. We acknowledge that it is of high relevance to state the definition of MCI used in each cohort, as well as the limitations that using different definitions would bring up. In this case, for both ADNI and EPAD, we specified MCI by a Clinical Dementia Rating = 0.5. We have made sure to state this in the main text:

Materials and methods, Page 25, lines 675-676: “MCI individuals, which were only included from ADNI and EPAD cohorts, were specified by a Clinical Dementia Rating = 0.5.”.

– It is inappropriate to report accuracy performance metrics after correction for brain-age bias (e.g., Table 3). The point of the correction process is not to improve model performance post hoc, but instead to improve the interpretability of the metrics by removing residual correlation with age. All performance metrics 'after bias correction' should be removed from the manuscript

We appreciate your concern regarding the reporting of accuracy performance metrics after correction for brain-age bias. We understand the point you raised regarding the purpose of the correction process, which is to improve the interpretability of the metrics by removing residual correlation with age. Therefore, we have removed all the performance metrics “after bias correction” from the manuscript.

– Figure 4. It was not obvious what the interactions related to. Of the 9 interactions reported, one is by sex, the other by age. Also, why were the points grouped and coloured by positive versus negative brain-age δ? Generally, this figure could be improved for the sake of clarity.

Thank you for your feedback, we appreciate your attention on Figure 4. To clarify, the interactions are related to the relationship between the different biomarkers and brain-age δ across the range of chronological age. The interaction reported by sex was a typo that we had not realized before. Thank you again for your attention on these details.

Regarding the question about the grouping and coloring of the points, the grouping in two different groups is only for visual purposes. We believed that by grouping the individuals by positive versus negative brain-age δ we would highlight the difference between individuals who have a brain that is biologically younger than their chronological age (positive δ) and those who have a brain that is biologically older (negative δ) and, in this way, it would be more clearly shown the interaction effects with age. We recognize that the explanation for this was not clearly stated. Following your recommendation, we have worked to improve the clarity of the figure and its legend to better communicate our results.

Results, page 20: “Figure 5. The associations of brain-age δ and (a) CSF NfL and (b) plasma NfL for all CU and, when available, MCI individuals. For visualization purposes, individuals were categorized into 2 groups according to their brain-age δ: “brain-age δ < 0” representing decelerated brain aging (blue); and “brain-age δ > 0” representing accelerated brain aging (red). Scatter plots representing the associations of CSF NfL, plasma NfL and WMH with age in individuals with brain-age δ > 0 and brain-age δ < 0. Each point depicts the value of the validation biomarkers of an individual and the solid lines indicate the regression line for each of the groups. The regression coefficients (β) and the corrected P-values are shown and were computed using a linear model adjusting for age and sex. Additionally, we also computed the “brain-age δ x sex” interaction term.”

– Why was the 'aging signature' included? Please provide stronger motivation for why this was relevant.

We thank the reviewer for this relevant question. Dickerson’s aging signature reflect those brain regions that undergo cortical atrophy in normal aging. Therefore, we were motivated to compute this simple measurement as a ‘control’ method to (1) show the superiority of brain-aging methods to more simple estimates of brain aging and (2) to compute a simple metric of longitudinal brain changes associated with aging. However, we agree that showing cross-sectional aging signatures in association with brain-age δ is not necessary for our aims. Therefore, in order to simplify our manuscript, we have now removed these associations.

– The analysis on the top and bottom 10% of participants were not well motivated. Why include this when regression models have already by used that assess 100% of the data, instead of 20%? It was not clear where these results were reported either, so I would recommend omitting this analysis.

Our goal with this secondary analysis was to examine the impact of extreme individuals on the overall relationships observed in our sample in the expectation that the observed effect would be small. Of course, this has the drawback that some of the data is discarded, but, in spite of this, the resulting analyses may be more sensitive. We acknowledge that the results of the analyses with extreme groups add little to the main ones and, therefore, they are only presented as supplementary material for future reference.

We recognize that the analysis of the top and bottom 10% may not have been clearly motivated and we apologize for any confusion it may have caused. We have also reported the results of this analysis more clearly in the Supplementary Information section.

Materials and methods, Page 30, lines 962-972: “Secondly, we wanted to identify the individuals whose predicted brain-age deviate the most from chronological aging, i.e., individuals with the highest positive or lowest negative brain-age deltas, to study the above-mentioned associations. The aim of this secondary analysis was to examine the impact of these extreme individuals on the overall relationships observed in our sample, in the expectation that the observed effect would be small. Even though this procedure implies that some of the data is discarded, we expected that the resulting analyses may be more sensitive. With this aim, we selected the individuals whose brain-age δ was included within the 10th and 90th percentile of the distribution for each independent cohort and studied the differences between these groups. The methodology and the results of this analysis can be found in Supplementary Appendix A.”

– Avoid drawing interpretations that males and females are different in cases where there is no statistical test of this hypothesis. Being non-identical does not necessarily mean that sex differences are statistically significant (e.g., Figure 1d).

We agree with the reviewer on this point and have, therefore, revised the relevant sections of the manuscript to ensure that all interpretations are based on the results of the statistical tests performed. We have included a table in Suplemmentary Information (Supplementary Table S8) with an exploratory analysis in which we have performed linear regressions for each region that was selected as more important in the predication (according to SHAP values) for which we included the interaction term age*sex.

– I recommend reporting confidence intervals whenever you report an effect size. This facilitates the comparison of effect sizes between tests.

We thank the reviewer for this recommendation and following it, we have now made sure to include the confidence intervals in the cases where we had not done it before due to lack of space. (e.g., Table 4).